# FAIRNESS AND ACCURACY UNDER DOMAIN GENERALIZATION

**Thai-Hoang Pham, Xueru Zhang, Ping Zhang**
The Ohio State University, Columbus, OH 43210, USA
`{pham.375,zhang.12807,zhang.10631}@osu.edu`

## ABSTRACT

As machine learning (ML) algorithms are increasingly used in high-stakes applications, concerns have arisen that they may be biased against certain social groups. Although many approaches have been proposed to make ML models fair, they typically rely on the assumption that data distributions in training and deployment are identical. Unfortunately, this is commonly violated in practice and a model that is fair during training may lead to an unexpected outcome during its deployment. Although the problem of designing robust ML models under dataset shifts has been widely studied, most existing works focus only on the transfer of accuracy. In this paper, we study the transfer of both fairness and accuracy under domain generalization where the data at test time may be sampled from *never-before-seen* domains. We first develop theoretical bounds on the unfairness and expected loss at deployment, and then derive sufficient conditions under which fairness and accuracy can be perfectly transferred via invariant representation learning. Guided by this, we design a learning algorithm such that fair ML models learned with training data still have high fairness and accuracy when deployment environments change. Experiments on real-world data validate the proposed algorithm. Model implementation is available at `https://github.com/pth1993/FATDM`.

## 1 INTRODUCTION

Machine learning (ML) algorithms trained with real-world data may have inherent bias and exhibit discrimination against certain social groups. To address the unfairness in ML, existing studies have proposed many fairness notions and developed approaches to learning models that satisfy these fairness notions. However, these works are based on an implicit assumption that the data distributions in training and deployment are the same, so that the fair models learned from training data can be deployed to make fair decisions on testing data. Unfortunately, this assumption is commonly violated in real-world applications such as healthcare e.g., it was shown that most US patient data for training ML models are from CA, MA, and NY, with almost no representation from the other 47 states (Kaushal et al., 2020). Because of the distribution shifts between training and deployment, a model that is accurate and fair during training may behave in an unexpected way and induce poor performance during deployment. Therefore, it is critical to account for distribution shifts and learn fair models that are robust to potential changes in deployment environments.

The problem of learning models under distribution shifts has been extensively studied in the literature and is typically referred to as domain adaptation/generalization, where the goal is to learn models on *source* domain(s) that can be generalized to a different (but related) *target* domain. Specifically, *domain adaptation* requires access to (unlabeled) data from the target domain at training time, and the learned model can only be used at a specific target domain. In contrast, *domain generalization* considers a more general setting when the target domain data are inaccessible during training; instead it assumes there exists a set of source domains based on which the learned model can be generalized to an unseen, novel target domain. For both problems, most studies focus only on the generalization of accuracy across domains without considering fairness, e.g., by theoretically examining the relations between accuracy at target and source domains (Mansour et al., 2008; 2009; Hoffman et al., 2018; Zhao et al., 2018; Phung et al., 2021; Deshmukh et al., 2019; Muandet et al., 2013; Blanchard et al., 2021; Albuquerque et al., 2019; Ye et al., 2021; Sicilia et al., 2021; Shui et al., 2022) or/and developing practical methods (Albuquerque et al., 2019; Zhao et al., 2020; Li et al., 2018a; Sun &

Saenko, 2016; Ganin et al., 2016; Ilse et al., 2020; Nguyen et al., 2021). To the best of our knowledge, only Chen et al. (2022); Singh et al. (2021); Coston et al. (2019); Rezaei et al. (2021); Oneto et al. (2019); Madras et al. (2018); Schumann et al. (2019); Yoon et al. (2020) considered the transfer of fairness across domains. However, all of them focused on domain adaptation, and many also imposed rather strong assumptions on distributional shifts (e.g., covariate shifts (Singh et al., 2021; Coston et al., 2019; Rezaei et al., 2021), demographic shift (Giguere et al., 2022), prior probability shift (Biswas & Mukherjee, 2021)) that may be violated in practice. Among them, most focused on empirically examining how fairness properties are affected under distributional shifts, whereas theoretical understandings are less studied (Schumann et al., 2019; Yoon et al., 2020). Details and more related works are in Appendix A.

In this paper, we study the transfer of both fairness and accuracy in **domain generalization** via invariant representation learning, where the data in target domain is *unknown* and *inaccessible* during training. A motivating example is shown in Figure 1. Specifically, we first establish a new theoretical framework that develops interpretable bounds on accuracy/fairness at a target domain under domain generalization, and then identify sufficient conditions under which fairness/accuracy can be perfectly transferred to an *unseen* target domain. Importantly, our theoretical bounds are fundamentally different from the existing bounds, compared to which ours are better connected with practical algorithmic design, i.e., our bounds are aligned with the objective of *adversarial learning-based* algorithms, a method that is widely used in domain generalization.

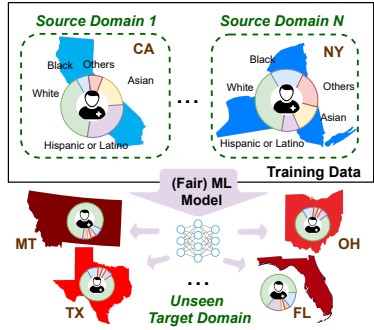

Figure 1: An example of domain generalization in healthcare: (fair) ML model trained with patient data in CA, NY, etc., can be deployed in other states by maintaining high accuracy/fairness.

Inspired by the theoretical findings, we propose *Fairness and Accuracy Transfer by Density Matching* (`FATDM`), a two-stage learning framework such that the representations and fair model learned with source domain data can be well-generalized to an unseen target domain. Last, we conduct the experiments on real-world data; the empirical results show that fair ML models trained with our method still attain a high accuracy and fairness when deployment environments differ from the training. Our main contributions and findings are summarized as follows:

- We consider the transfer of both *accuracy* and *fairness* in domain generalization. To the best of our knowledge, this is the first work studying domain generalization with fairness consideration.
- We develop *upper* bounds for expected loss (Thm. 1) and unfairness (Thm. 3) in target domains. Notably, our bounds are significantly different from the existing bounds as discussed in Appendix A. We also develop a *lower* bound for expected loss (Thm. 2); it indicates an inherent tradeoff of the existing methods which learn marginal invariant representations for domain generalization.
- We identify sufficient conditions under which fairness and accuracy can be *perfectly* transferred from source domains to target domains using invariant representation learning (Thm. 4).
- We propose a two-stage training framework (i.e., based on Thm. 5) that learns models in source domains (Sec. 4), which can generalize both accuracy and fairness to target domain.
- We conduct experiments on real-world data to validate the effectiveness of the proposed method.

## 2 PROBLEM FORMULATION

**Notations.** Let $\mathcal{X}, \mathcal{A}$, and $\mathcal{Y}$ denote the space of features, sensitive attribute (distinguishing different groups, e.g., race/gender), and label, respectively. Let $\mathcal{Z}$ be the representation space induced from $\mathcal{X}$ by representation mapping $g : \mathcal{X} \to \mathcal{Z}$. We use $X, A, Y, Z$ to denote random variables that take values in $\mathcal{X}, \mathcal{A}, \mathcal{Y}, \mathcal{Z}$ and $x, a, y, z$ the realizations. A *domain* $D$ is specified by distribution $P_D : \mathcal{X} \times \mathcal{A} \times \mathcal{Y} \to [0, 1]$ and labeling function $f_D : \mathcal{X} \to \mathcal{Y}^\Delta$, where $\Delta$ is a probability simplex over $\mathcal{Y}$. Similarly, let $h_D : \mathcal{Z} \to \mathcal{Y}^\Delta$ be a labeling function from representation space for domain $D$. Note that $f_D, h_D, g$ are stochastic functions and $f_D = h_D \circ g$.[1] For simplicity, we use $P_D^V$ (or $P_D^{V|U}$) to denote the induced marginal (or conditional) distributions of variable $V$ (given $U$) in domain $D$.

---

[1]The deterministic labeling function is a special case when it follows Dirac delta distribution in $\Delta$.

**Error metric.** Consider hypothesis $\widehat{f} = \widehat{h} \circ g : \mathcal{X} \to \mathcal{Y}^\Delta$, where $\widehat{h} : \mathcal{Z} \to \mathcal{Y}^\Delta$ is the hypothesis directly used in representation space. Denote $\widehat{f}(x)_y$ as the element on $y$-th dimension which predicts the probability that label $Y = y$ given $X = x$. Then the expected error of $\widehat{f}$ in domain $D$ is defined as $\epsilon_D^{Acc}(\widehat{f}) = \mathbb{E}_D[\mathcal{L}(\widehat{f}(X), Y)]$ for some loss function $\mathcal{L} : \mathcal{Y}^\Delta \times \mathcal{Y} \to \mathbb{R}_+$ (e.g., 0-1 loss, cross-entropy loss). Similarly, define the expected error of $\widehat{h}$ in representation space as $\epsilon_D^{Acc}(\widehat{h}) = \mathbb{E}_D[\mathcal{L}(\widehat{h}(Z), Y)]$. Note that most existing works (Albuquerque et al., 2019; Zhao et al., 2018) focus on optimizing $\epsilon_D^{Acc}(\widehat{h})$, while our goal is to attain high accuracy in input space, i.e., low $\epsilon_D^{Acc}(\widehat{f})$.

**Unfairness metric.** We focus on group fairness notions (Makhlouf et al., 2021) that require certain statistical measures to be equalized across different groups; many of them can be formulated as (conditional) independence statements between random variables $\widehat{f}(X), A, Y$, e.g., *demographic parity* ($\widehat{f}(X) \perp A$: the likelihood of a positive outcome is the same across different groups) (Dwork et al., 2012) , *equalized odds* ($\widehat{f}(X) \perp A|Y$: true positive rate (TPR) and false positive rate (FPR) are the same across different groups), *equal opportunity* ($\widehat{f}(X) \perp A|Y = 1$ when $\mathcal{Y} = \{0, 1\}$: TPR is the same across different groups) (Hardt et al., 2016). In the paper, we will present the results under *equalized odds* (EO) fairness with binary $\mathcal{Y} = \{0, 1\}$ and $\mathcal{A} = \{0, 1\}$, while all the results (e.g., methods, analysis) can be generalized to multi-class, multi-protected attributes, and other fairness notions. Given hypothesis $\widehat{f} = \widehat{h} \circ g : \mathcal{X} \to \mathcal{Y}^\Delta$, the violation of EO in domain $D$ can be measured as $\epsilon_D^{EO}(\widehat{f}) = \sum_{y \in \mathcal{Y}} \mathcal{D}\left(P_D^{\widehat{f}(X)_1|Y=y,A=0}||P_D^{\widehat{f}(X)_1|Y=y,A=1}\right)$ for some distance metric $\mathcal{D}(\cdot||\cdot)$.

**Problem setup.** Consider a problem of domain generalization where a learning algorithm has access to data $\{(x_k, a_k, y_k, d_k)\}_{k=1}^m$ sampled from a set of $N$ source domains $\{D_i^S\}_{i \in [N]}$, where $d_k$ is the domain label and $[N] = \{1, \cdots, N\}$. Our goal is to learn a representation mapping $g : \mathcal{X} \to \mathcal{Z}$ and a fair model $\widehat{h} : \mathcal{Z} \to \mathcal{Y}^\Delta$ trained on source domains such that the model $\widehat{f} = \widehat{h} \circ g$ can be generalized to an *unseen* target domain $D^T$ in terms of both accuracy and fairness. Specifically, we investigate under what conditions and by what algorithms we can guarantee that attaining high accuracy and fairness at source domains $\{D_i^S\}_{i=1}^N$ implies small $\epsilon_{D^T}^{Acc}(\widehat{f})$ and $\epsilon_{D^T}^{EO}(\widehat{f})$ at unknown target domain.

## 3   THEORETICAL RESULTS

In this section, we present the results on the transfer of accuracy/fairness under domain generalization via domain-invariant learning (proofs are shown in Appendix E). We first examine that for *any* model $\widehat{h} : \mathcal{Z} \to \mathcal{Y}^\Delta$ and *any* representation mapping $g : \mathcal{X} \to \mathcal{Z}$, how the accuracy/fairness attained at source domains $\{D_i^S\}_{i=1}^N$ can be affected when $\widehat{f} = \widehat{h} \circ g$ is deployed at *any* target domain $D^T$. Specifically, we can bound the error and unfairness at *any* target domain based on source domains. Before presenting the results, we first introduce the discrepancy measure used for measuring the dissimilarity between domains.

**Discrepancy measure.** We adopt Jensen-Shannon (JS) distance (Endres & Schindelin, 2003) to measure the dissimilarity between two distributions. Formally, JS distance between distributions $P$ and $P'$ is defined as

$$d_{JS}(P, P') := \sqrt{\mathcal{D}_{JS}(P||P')},$$

where $\mathcal{D}_{JS}(P||P') := \frac{1}{2}\mathcal{D}_{KL}(P||\frac{P+P'}{2}) + \frac{1}{2}\mathcal{D}_{KL}(P'||\frac{P+P'}{2})$ is JS divergence defined based on Kullback–Leibler (KL) divergence $\mathcal{D}_{KL}(\cdot||\cdot)$. Note that unlike KL divergence, JS divergence is symmetric and bounded: $0 \le \mathcal{D}_{JS}(P||P') \le 1$.

While different discrepancy measures such as $\mathcal{H}$ and $\mathcal{H}\Delta\mathcal{H}$ divergences (Ben-David et al., 2010) (i.e., definitions are given in Appendix A) were used in prior works, we particularly consider JS distance because (1) it is aligned with training objective for discriminator in generative adversarial networks (GAN) (Goodfellow et al., 2014), and many existing methods (Ganin et al., 2016; Albuquerque et al., 2019) for invariant representation learning are built based on GAN framework; (2) $\mathcal{H}$ and $\mathcal{H}\Delta\mathcal{H}$ divergences are limited to settings where the labeling functions $f_D$ are deterministic (Ben-David et al., 2010; Albuquerque et al., 2019; Zhao et al., 2018). In contrast, our bounds admit the stochastic labeling functions. The limitations of other discrepancy measures and existing bounds are discussed in detail in Appendix A.

**Theorem 1 (Upper bound: accuracy)** *For any hypothesis $\widehat{h} : \mathcal{Z} \to \mathcal{Y}^\Delta$, any representation mapping $g : \mathcal{X} \to \mathcal{Z}$, and any loss function $\mathcal{L} : \mathcal{Y}^\Delta \times \mathcal{Y} \to \mathbb{R}_+$ that is upper bounded by $C$, the expected error of $\widehat{f} = \widehat{h} \circ g : \mathcal{X} \to \mathcal{Y}^\Delta$ at any unseen target domain $D^T$ is upper bounded:*[2]

$$\epsilon_{D^T}^{Acc}\left(\widehat{f}\right) \leq \underbrace{\frac{1}{N}\sum_{i=1}^{N} \epsilon_{D_i^S}^{Acc}\left(\widehat{f}\right)}_{\textit{term (i)}} + \underbrace{\sqrt{2}C \min_{i \in [N]} d_{JS}\left(P_{D^T}^{X,Y}, P_{D_i^S}^{X,Y}\right)}_{\textit{term (ii)}} + \underbrace{\sqrt{2}C \max_{i,j \in [N]} d_{JS}\left(P_{D_i^S}^{Z,Y}, P_{D_j^S}^{Z,Y}\right)}_{\textit{term (iii)}} \quad (1)$$

The upper bound in Eq. (1) are interpretable and have three terms: **term (i)** is the averaged error of source domains in input space; **term (ii)** is the discrepancy between the target domain and the source domains in input space; **term (iii)** is the discrepancy between the source domains in representation space.[3] It provides guidance on learning the proper representation mapping $g : \mathcal{X} \to \mathcal{Z}$: to ensure small error at target domain $\epsilon_{D^T}^{Acc}(\widehat{f})$, we shall learn representations such that the upper bound of $\epsilon_{D^T}^{Acc}(\widehat{f})$ is minimized. Because **term (ii)** depends on the unknown target domain $D^T$ and it's evaluated in input space $\mathcal{X} \times \mathcal{Y}$, it is fixed and is out of control during training, we can only focus on **term (i)** and **term (iii)**, i.e., learn representations $Z$ such that errors at source domains $\epsilon_{D_i^S}^{Acc}(\widehat{f})$ and the discrepancy between source domains in the representation space $d_{JS}(P_{D_i^S}^{Z,Y}, P_{D_j^S}^{Z,Y})$ are minimized.

**Corollary 1.1** $\forall i, j$, *JS distance between $P_{D_i^S}^{Z,Y}$ and $P_{D_j^S}^{Z,Y}$ in Eq.* (1) *can be decomposed:*

$$d_{JS}\left(P_{D_i^S}^{Z,Y}, P_{D_j^S}^{Z,Y}\right) = d_{JS}\left(P_{D_i^S}^{Y}, P_{D_j^S}^{Y}\right) + \sqrt{2\mathbb{E}_{y \sim P_{D_{i,j}^S}^{Y}}\left[d_{JS}\left(P_{D_i^S}^{Z|Y}, P_{D_j^S}^{Z|Y}\right)^2\right]}$$

*where $P_{D_{i,j}^S}^{Y} = \frac{1}{2}\left(P_{D_i^S}^{Y} + P_{D_j^S}^{Y}\right)$.*

Our algorithm in Sec. 4 is designed based on above decomposition: because $P_{D_i^S}^{Y}$ solely depends on source domain $D_i^S$, we learn representations by minimizing $d_{JS}(P_{D_i^S}^{Z|Y}, P_{D_j^S}^{Z|Y}), \forall i, j$. Combining Thm. 1 and Corollary 1.1, to ensure high accuracy at unseen target domain $D^T$, we learn the representation mapping $g$ and model $\widehat{h}$ such that $P_{D_i^S}^{Z|Y}$ is invariant across source domains, and meanwhile $\widehat{f} = \widehat{h} \circ g$ attains high accuracy at source domains.

Note that unlike our method, many existing works (Phung et al., 2021; Albuquerque et al., 2019; Ganin et al., 2016) suggest that to ensure high accuracy in domain generalization, representation mapping $g$ should be learned such that $P_{D_i^S}^{Z}$ is same across domains, i.e., small $d_{JS}(P_{D_i^S}^{Z}, P_{D^T}^{Z})$. However, we show that the domain-invariant $P_{D_i^S}^{Z}$ may adversely increase the error at target domain, as indicated in the Thm. 2 below.

**Theorem 2 (Lower bound: accuracy)** *Suppose $\mathcal{L}(\widehat{f}(x), y) = \sum_{\widehat{y} \in \mathcal{Y}} \widehat{f}(x)_{\widehat{y}} L(\widehat{y}, y)$ where function $L : \mathcal{Y} \times \mathcal{Y} \to \mathbb{R}_+$ is lower bounded by $c$ when $\widehat{y} \neq y$, and is 0 when $\widehat{y} = y$. If $d_{JS}(P_{D_i^S}^{Y}, P_{D^T}^{Y}) \geq d_{JS}(P_{D_i^S}^{Z}, P_{D^T}^{Z})$, the expected error of $\widehat{f}$ at source and target domains is lower bounded:*

$$\frac{1}{N}\sum_{i=1}^{N} \epsilon_{D_i^S}^{Acc}(\widehat{f}) + \epsilon_{D^T}^{Acc}(\widehat{f}) \geq \frac{c}{4|\mathcal{Y}|N}\sum_{i=1}^{N}\left(d_{JS}(P_{D_i^S}^{Y}, P_{D^T}^{Y}) - d_{JS}(P_{D_i^S}^{Z}, P_{D^T}^{Z})\right)^4. \quad (2)$$

The above lower bound shows an inherent trade-off of approaches that minimize $d_{JS}(P_{D_i^S}^{Z}, P_{D^T}^{Z})$ when learning the representations. Specifically, with the domain-invariant $P_{D_i^S}^{Z}$, the right hand side of Eq. (2) may increase, resulting in an increased error at target domain $\epsilon_{D^T}^{Acc}(\widehat{f})$.

---

[2]The condition on the bounded loss is mild and can be satisfied by many loss functions. For example, cross-entropy loss can be bounded by modifying the softmax output from $(p_1, p_2, \cdots, p_{|\mathcal{Y}|})$ to $(\hat{p}_1, \hat{p}_2, \cdots, \hat{p}_{|\mathcal{Y}|})$, where $\hat{p}_i = p_i(1 - \exp(-C)|\mathcal{Y}|) + \exp(-C), \forall i \in |\mathcal{Y}|$.

[3]In fact, a tighter upper bound for the loss at target domain can be established using strong data processing inequality (Polyanskiy & Wu, 2017), as detailed in Appendix D

Similar to the loss, the unfairness at target domain can also be upper bounded, as presented in Thm. 3.

**Theorem 3 (Upper bound: fairness)** *Consider a special case where the unfairness measure is defined as the distance between means of two distributions:*

$$\epsilon_D^{EO}(\widehat{f}) = \sum_{y\in\{0,1\}} \left| \mathbb{E}_D\left[\widehat{f}(X)_1|Y=y, A=0\right] - \mathbb{E}_D\left[\widehat{f}(X)_1|Y=y, A=1\right] \right|,$$

*then the unfairness at any unseen target domain $D^T$ is upper bounded:*

$$
\begin{aligned}
\epsilon_{D^T}^{EO}\left(\widehat{f}\right) \leq \ & \frac{1}{N}\sum_{i=1}^{N}\epsilon_{D_i^S}^{EO}\left(\widehat{f}\right) + \sqrt{2}\min_{i\in[N]}\sum_{y\in\{0,1\}}\sum_{a\in\{0,1\}}d_{JS}\left(P_{D^T}^{X|Y=y,A=a}, P_{D_i^S}^{X|Y=y,A=a}\right) \\
& + \sqrt{2}\max_{i,j\in[N]}\sum_{y\in\{0,1\}}\sum_{a\in\{0,1\}}d_{JS}\left(P_{D_i^S}^{Z|Y=y,A=a}, P_{D_j^S}^{Z|Y=y,A=a}\right)
\end{aligned}
$$

Similar to Thm. 1, the upper bound in Thm. 3 also has three terms and the second term is out of control during training because it depends on the unseen target domain and is defined in input space. Therefore, to maintain fairness at target domain $D^T$, we learn the representation mapping $g$ and model $\widehat{h}$ such that $P_{D_i^S}^{Z|Y,A}$ is invariant across source domains, and meanwhile $\widehat{f} = \widehat{h} \circ g$ attains high fairness at source domains.

The results above characterize the relations between accuracy/fairness at any target and source domains under any representation mapping $g$ and model $\widehat{h}$. Next, we identify conditions under which the accuracy/fairness attained at sources can be perfectly transferred to a target domain.

**Theorem 4 (Sufficient condition for perfect transfer)** *Consider $N$ source domains $\{D_i^S\}_{i=1}^N$ and an unseen target domain $D^T$. Define set $\Lambda = \{D^t : D^t = \sum_{i=1}^N \pi_i D_i^S, \{\pi_i\} \in \Delta_{N-1}\}$.*

1. *(**Transfer of fairness**) $\forall D^T \in \Lambda$, if $g$ is the mapping under which $P_{D_i^S}^{Z|Y,A}$ is the same across all source domains, then $\epsilon_{D_i^S}^{EO}(\widehat{h}) = \epsilon_{D^T}^{EO}(\widehat{h}) = \epsilon_{D_i^S}^{EO}(\widehat{f}) = \epsilon_{D^T}^{EO}(\widehat{f}), \ \forall i$.*

2. *(**Transfer of accuracy**) $\forall D^T \in \Lambda$, if $P_{D_i^S}^{Y}$ is the same and if $g$ is the mapping under which $P_{D_i^S}^{Z|Y}$ is the same across all source domains, then $\epsilon_{D_i^S}^{ACC}(\widehat{h}) = \epsilon_{D^T}^{ACC}(\widehat{h}) = \epsilon_{D_i^S}^{ACC}(\widehat{f}) = \epsilon_{D^T}^{ACC}(\widehat{f}), \ \forall i$.*

Thm. 4 indicates the possibility of attaining the perfect transfer of accuracy/fairness and examples of such representation mappings are provided. Note that these results are consistent with Thm. 1 and Thm. 3, which also suggest learning domain-invariant representations $P_{D_i^S}^{Z|Y}$ and $P_{D_i^S}^{Z|Y,A}$.

## 4 PROPOSED ALGORITHM

The accuracy and fairness upper bounds in Sec. 3 shed light on designing robust ML model that can preserve high accuracy and fairness on unseen target domains. Specifically, the model consists of representation mapping $g : \mathcal{X} \to \mathcal{Z}$ and classifier $\widehat{h} : \mathcal{Z} \to \mathcal{Y}$ such that (1) the prediction errors and unfairness of $\widehat{f} = \widehat{h} \circ g$ on source domains are minimized; and (2) the discrepancy of learned conditional representations (i.e, $P_{D_i^S}^{Z|Y}$ and $P_{D_i^S}^{Z|Y,A}$) among source domains is minimized. That is,

Table 1: Usages of terms in Eq. (3) to guarantee the fairness and accuracy in target domain.

| Loss terms | Usages |
|---|---|
| $\mathcal{L}_{cls}$ | Mimimize $\epsilon_{D_i^S}^{ACC}$ |
| $\mathcal{L}_{fair}$ | Mimimize $\epsilon_{D_i^S}^{EO}$ |
| $\mathcal{L}_{inv}$ | Minimize $d_{JS}\left(P_{D_i^S}^{Z|Y}, P_{D_j^S}^{Z|Y}\right)$ and $d_{JS}\left(P_{D_i^S}^{Z|Y,A}, P_{D_j^S}^{Z|Y,A}\right)$ |
| $\mathcal{L}_{cls} + \mathcal{L}_{fair} + \mathcal{L}_{inv}$ | Mimimize $\epsilon_{D^T}^{ACC}$ and $\epsilon_{D^T}^{EO}$ |

$$\min_{g,\widehat{h}} \mathcal{L}_{cls}(g,\widehat{h}) + \omega\mathcal{L}_{fair}(g,\widehat{h}) + \gamma\mathcal{L}_{inv}(g) \tag{3}$$

where $\mathcal{L}_{cls}$, $\mathcal{L}_{fair}$, and $\mathcal{L}_{inv}$ are expected losses that penalize incorrect classification, unfairness, and discrepancy among source domains. Hyper-parameters $\omega > 0$ and $\gamma > 0$ control the accuracy-fairness trade-off and accuracy-invariant representation trade-off, respectively. The usages of these three losses are summarized in Table 1.

**Adversarial learning framework (Goodfellow et al., 2014).** $\mathcal{L}_{inv}$ in Eq. (3) can be optimized directly with adversarial learning. This is because the training objective of the discriminator in GAN is aligned with our goal of minimizing JS distance between $P_{D_i^S}^{Z|Y}$ (or $P_{D_i^S}^{Z|Y,A}$) among source domains, as mentioned in Sec. 3. Specifically, define a set of discriminators $\mathcal{K} = \{k_y : y \in \mathcal{Y}\} \cup \{k_{y,a} : y \in \mathcal{Y}, a \in \mathcal{A}\}$; each discriminator $k_y$ (resp. $k_{y,a}$) aims to distinguish whether a sample with label $y$ (resp. label $y$ and sensitive attribute $a$) comes from a particular domain (i.e., maximize $\mathcal{L}_{inv}$). The representation mapping $g$ should be learned to increase the error of discriminators (i.e., minimize $\mathcal{L}_{inv}$). Therefore, the model and discriminators can be trained simultaneously by playing a two-player minimax game (i.e., $\min_g \max_{\mathcal{K}} \mathcal{L}_{inv}(g)$). Combine with the objective of minimizing prediction error and unfairness (i.e., $\min_{g,\widehat{h}} \mathcal{L}_{cls}(g, \widehat{h}) + \omega \mathcal{L}_{fair}(g, \widehat{h})$), the overall learning objective is:

$$\min_{g,\widehat{h}} \max_{\mathcal{K}} \mathcal{L}_{cls}(g, \widehat{h}) + \omega \mathcal{L}_{fair}(g, \widehat{h}) + \gamma \mathcal{L}_{inv}(g) \tag{4}$$

However, the above adversarial learning framework for learning domain-invariant representation may not work well when $|\mathcal{Y} \times \mathcal{A}|$ is large: as the label space and sensitive attribute space get larger, the number of discriminators to be learned increases and the training can be highly unstable. A naive solution to tackling this issue is to use one discriminator $\forall y \in \mathcal{Y}, a \in \mathcal{A}$. However, this would result in the reduced mutual information between representations and label/sensitive attribute, which may hurt the accuracy. We thus propose another approach to learn the domain-invariant representations.

**Proposed solution to learning invariant representations.** For any domain $D$, we have:

$$P_D^{Z|y} = \int_{\mathcal{X}} P_D^{Z,x|y} dx = \int_{\mathcal{X}} P^{Z|x} P_D^{x|y} dx$$

$$P_D^{Z|y,a} = \int_{\mathcal{X}} P_D^{Z,x|y,a} dx = \int_{\mathcal{X}} P^{Z|x} P_D^{x|y,a} dx$$

where $P^{Z|x}$ is domain-independent so we drop $D$ in subscript. Given any two source domains $D_i^S$ and $D_j^S$, in general $P_{D_i^S}^{X|y} \neq P_{D_j^S}^{X|y}$ and $P_{D_i^S}^{X|y,a} \neq P_{D_j^S}^{X|y,a}$ so that it is nontrivial to achieve domain-invariant representations $P_{D_i^S}^{Z|y} = P_{D_j^S}^{Z|y}$ and $P_{D_i^S}^{Z|y,a} = P_{D_j^S}^{Z|y,a}$. How-

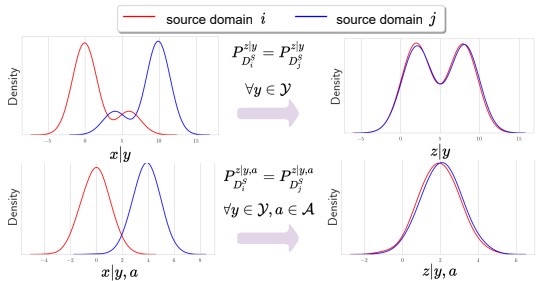

Figure 2: 1D illustration of domain-invariant representation. To transfer accuracy and fairness to target domains, we need to find representation $z$ such that $P_{D_i^S}^{z|y}$ and $P_{D_i^S}^{z|y,a}$ are domain-invariant.

ever, if there exist invertible functions $m_{i,j}^y : \mathcal{X} \rightarrow \mathcal{X}$ and $m_{i,j}^{y,a} : \mathcal{X} \rightarrow \mathcal{X}$ that can match the density functions of $X$ from $D_i^S$ to $D_j^S$ such that $P_{D_i^S}^{X|y} = P_{D_j^S}^{m_{i,j}^y(X)|y}$ and $P_{D_i^S}^{X|y,a} = P_{D_j^S}^{m_{i,j}^{y,a}(X)|y,a}$, and if we can find the representation $Z$ such that $P_{D_i^S}^{Z|x} = P_{D_j^S}^{Z|m_{i,j}^y(x)}$ and $P_{D_i^S}^{Z|x} = P_{D_j^S}^{Z|m_{i,j}^{y,a}(x)}$, then $\forall y \in \mathcal{Y}, a \in \mathcal{A}$, we have:

$$P_{D_i^S}^{Z|y} = \int_{\mathcal{X}} P^{Z|x} P_{D_i^S}^{x|y} dx = \int_{\mathcal{X}} P^{Z|x'} P_{D_j^S}^{x'|y} dx' = P_{D_j^S}^{Z|y}$$

$$P_{D_i^S}^{Z|y,a} = \int_{\mathcal{X}} P^{Z|x} P_{D_i^S}^{X|y,a} dx = \int_{\mathcal{X}} P^{Z|x''} P_{D_j^S}^{x''|y,a} dx'' = P_{D_j^S}^{Z|y,a}$$

where $x' = m_{i,j}^y(x)$ and $x'' = m_{i,j}^{y,a}(x)$. This observation suggests that to minimize the discrepancy of representation distributions among source domains, we can first find the density mapping functions $m_{i,j}^y$ and $m_{i,j}^{y,a}, \forall y, a, i, j$, and then minimize the discrepancies between $P^{Z|x}, P^{Z|x'}$, and $P^{Z|x''}$, $\forall x$. This is formally shown in Thm. 5 below.

**Theorem 5** *If there exist invertible mappings $m_{i,j}^y$ and $m_{i,j}^{y,a}$ such that $P_{D_i^S}^{X|y} = P_{D_j^S}^{m_{i,j}^y(X)|y}$ and $P_{D_i^S}^{X|y,a} = P_{D_j^S}^{m_{i,j}^{y,a}(X)|y,a}, \forall y, a, i, j$, and if the representation mapping are in the form of $g := P^{Z|x} = \mathcal{N}(\mu(x), \sigma^2 I_d)$, where $\mu(x)$ is the function of $x$ and $d$ is the dimension of the representation space $\mathcal{Z}$, then minimizing $d_{JS}\left(P_{D_i^S}^{Z|y}, P_{D_j^S}^{Z|y}\right)$ and $d_{JS}\left(P_{D_i^S}^{Z|y,a}, P_{D_j^S}^{Z|y,a}\right)$ can be reduced to minimizing $\left\|\mu(x) - \mu\left(m_{i,j}^y(x)\right)\right\|_2$ and $\left\|\mu(x) - \mu\left(m_{i,j}^{y,a}(x)\right)\right\|_2$, respectively.*

Based on Thm. 5, we propose a two-stage learning approach FATDM, as stated below.

**Remark 1 (Fairness and Accuracy Transfer by Density Matching (FATDM))** *Given the existence of density matching functions $m_{i,j}^y$ and $m_{i,j}^{y,a}$, and representation mapping $g := \mathcal{N}(\mu(x), \sigma^2 I_d)$, domain-invariant representations can be learned via a two-stage process: (i) finding these mapping functions $m_{i,j}^y$ and $m_{i,j}^{y,a}$; (ii) minimizing the mean squared errors between $\mu(x)$ and $\mu\left(m_{i,j}^y(x)\right)$, and $\mu(x)$ and $\mu\left(m_{i,j}^{y,a}(x)\right), \forall i,j \in [N], x \in \mathcal{X}, y \in \mathcal{Y}, a \in \mathcal{A}.$*

***Stage 1: learning mapping functions $m_{i,j}^y$ and $m_{i,j}^{y,a}$ across source domains.*** Many approaches can be leveraged to estimate $m_{i,j}^y$ and $m_{i,j}^{y,a}$ from data. In our study, we adopt StarGAN (Choi et al., 2018) and CycleGAN (Zhu et al., 2017) as examples; both frameworks are widely used in multi-domain image-to-image translation and can be leveraged. In our algorithm, we independently train two translation models DensityMatch$^Y$ and DensityMatch$^{Y,A}$ using StarGAN or CycleGAN, with each used for learning $\{m_{i,j}^y\}_{y \in \mathcal{Y}, i,j \in [N]}$ and $\{m_{i,j}^{y,a}\}_{y \in \mathcal{Y}, a \in \mathcal{A}, i,j \in [N]}$, respectively.

Specifically, DensityMatch$^Y$ (or DensityMatch$^{Y,A}$) consists of a *generator* $\mathsf{G} : \mathcal{X} \times [N] \times [N] \to \mathcal{X}$ and a *discriminator* $\mathsf{D} : \mathcal{X} \to [N] \times \{0,1\}$. The generator takes in real image $x$ and a pair of domain labels $i, j$ as input and generates a fake image; the discriminator aims to predict the domain label of the image generated by the generator and distinguish whether it is fake or real. G and D are learned simultaneously by solving the minimax game, and their loss functions are specified in Appendix B. When the training is completed, we obtain two optimal generators from DensityMatch$^Y$ and DensityMatch$^{Y,A}$, denoted as $\mathsf{G}^Y$ and $\mathsf{G}^{Y,A}$. We shall use $\mathsf{G}^Y(\cdot, i, j)$ (resp. $\mathsf{G}^{Y,A}(\cdot, i, j)$) directly as the density mapping function $\{m_{i,j}^y(\cdot)\}_{y \in \mathcal{Y}}$ (resp. $\{m_{i,j}^{y,a}(\cdot)\}_{y \in \mathcal{Y}, a \in \mathcal{A}}$).

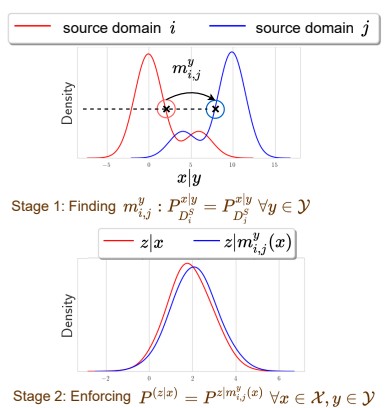

Stage 1: Finding $m_{i,j}^y$ : $P_{D_i^s}^{x|y} = P_{D_j^s}^{x|y} \ \forall y \in \mathcal{Y}$

Stage 2: Enforcing $P^{z|x} = P^{z|m_{i,j}^y(x)} \ \forall x \in \mathcal{X}, y \in \mathcal{Y}$

Figure 3: FATDM: two-stage training

***Stage 2: learning domain-invariant representation.*** Given $\mathsf{G}^Y$ and $\mathsf{G}^{Y,A}$ learned in stage 1, we are ready to learn the invariant representation $Z$ by finding $g : \mathcal{X} \to \mathcal{Z}$ such that $g := \mathcal{N}(\mu(x), \ \sigma^2 I_d)$ and minimizes the following:

$$\mathcal{L}_{inv} = \mathbb{E}_{d,d',d'' \sim \{D_i^S\}_{i \in [N]}} \left[ \mathcal{L}_{mse}(\mu(X), \mu(X')) + \mathcal{L}_{mse}(\mu(X), \mu(X'')) \right] \quad (5)$$

where $d, d', d''$ are domain labels sampled from source domains, $X$ is features sampled from domain $d$, $X' = \mathsf{G}^Y(X, d, d')$, $X'' = \mathsf{G}^{Y,A}(X, d, d'')$, $\mathcal{L}_{mse}$ is mean squared error. The pseudo-code of our proposed model (FATDM) is in Algorithm 1. The detailed architecture of FATDM is in Appendix B.

---

**Algorithm 1:** Fairness and Accuracy Transfer by Density Matching (FATDM)

---

**Input:** Training dataset $\mathcal{D}^{train}$ from $N$ source domains $\{D_i^S\}_{i=1}^N$

**Output:** representation mapping $g$, classifier $\widehat{h}$, density matching functions $\mathsf{G}^Y, \mathsf{G}^{Y,A}$

1 **Procedure** `Density_Matching`($\mathcal{D}^{train}$)

    /* Procedure for training $\mathsf{G}^Y$ is similar but not presented */

2     **while** *training DensityMatch$^{Y,A}$ is not end* **do**

3         Sample $y \sim \mathcal{Y}, a \sim \mathcal{A}$ and data batch $\mathcal{B} = \{x_k, d_k | a_k = a, y_k = y\}_{k=1}^{|\mathcal{B}|}$ from $\mathcal{D}^{train}$ ;

4         Update $\mathsf{G}^{Y,A}$ based on the objectives of the minimax game (Appendix B).

5 **Procedure** `Invariant_Representation_Learning`($\mathcal{D}^{train}, \mathsf{G}^Y, \mathsf{G}^{Y,A}$)

6     **while** *training FATDM is not end* **do**

7         Sample data batch $\mathcal{B} = \{x_k, a_k, y_k, d_k\}_{k=1}^{|\mathcal{B}|}$ from $\mathcal{D}^{train}$ ;

8         Sample lists of domain labels $\{d_i'\}_{k=1}^{|\mathcal{B}|}$ and $\{d_i''\}_{k=1}^{|\mathcal{B}|}$;

9         Generate sets of artificial images $\{x_k'\}_{k=1}^{|\mathcal{B}|}$ and $\{x_k''\}_{k=1}^{|\mathcal{B}|}$ by $\mathsf{G}^Y$ and $\mathsf{G}^{Y,A}$ ;

10         Update $g, \hat{h}$ by optimizing Eq. (3) with $\mathcal{L}_{inv}$ defined in Eq. (5).

---

**Remark 2 (Summary of theoretical results and proposed algorithm)** *Thm. 1 and Thm. 3 suggest a way to ensure high accuracy and fairness in target domain: by minimizing the source error $\epsilon_{D_i^s}^{Acc}$ (i.e., $\mathcal{L}_{cls}$ in Eq. (3)), the source unfairness $\epsilon_{D_i^s}^{EO}$ (i.e., $\mathcal{L}_{fair}$ in Eq. (3)), and the discrepancies between source domains $d_{JS}\left(P_{D_i^S}^{Z|Y=y}, P_{D_j^S}^{Z|Y=y}\right)$ and $d_{JS}\left(P_{D_i^S}^{Z|Y=y,A=a}, P_{D_j^S}^{Z|Y=y,A=a}\right)$ (i.e., $\mathcal{L}_{inv}$ in Eq. (3)). The*

*common way to optimize Eq. (3) using adversarial learning (Eq. (4)) is not stable when $|\mathcal{Y} \times \mathcal{A}|$ is large. Thm. 5 states that instead of using adversarial learning, Eq. (3) can be optimized via 2-stage learning: (i) find mappings $m_{i,j}^{y}$ and $m_{i,j}^{y,a}$ ( `Density_Matching` in Alg. 1) and (ii) minimize Eq. (3) with $\mathcal{L}_{inv}$ defined in Eq. (5)( `Invariant_Representation_Learning` in Alg. 1).*

## 5 EXPERIMENTS

We conduct experiments on MIMIC-CXR database (Johnson et al., 2019), which includes 377,110 chest X-ray images associated with 227,827 imaging studies about 14 diseases performed at the Beth Israel Deaconess Medical Center. Importantly, these images are linked with MIMIC-IV database (Johnson et al., 2021) which includes patients' information such as age, and race; these can serve as sensitive attributes for measuring the unfairness. Based on MIMIC-CXR and MIMIC-IV data, we construct two datasets on two diseases:

- *Cardiomegaly disease:* we first extract all images related to Cardiomegaly disease, and the corresponding labels (i.e., positive/negative) and sensitive attributes (i.e., male/female); then we partition the data into four domain-specific datasets based on age (i.e., $[18, 40), [40, 60), [60, 80), [80, 100)$). We consider age as domain label because it captures the real scenario that there are distribution shifts across patients with different ages.
- *Edema disease:* we extract all images related to Edema disease, and corresponding labels (i.e., positive/negative) and sensitive attributes (i.e., age with ranges $[18, 40), [40, 60), [60, 80), [80, 100)$). Unlike Cardiomegaly data, we construct the dataset for each domain by first sampling images from Edema data followed by $\theta$ degree counter-clockwise rotation, where $\theta \in \{0°, 15°, 30°, 45°, 60°\}$. We consider rotation degree as domain label to model the scenario where there is rotational misalignment among images collected from different devices.

Next, we focus on Cardiomegaly disease and the results for Edema disease are shown in Appendix C.

**Baselines.** We compare our method (i.e., `FATDM-StarGAN` and `FATDM-CycleGAN`) with existing methods for domain generalization, including empirical risk minimization, domain invariant representation learning, and distributionally robust optimization, as detailed below.

- Empirical risk minimization (`ERM`): The baseline that considers all source domains as one domain.
- Domain invariant representation learning: Method that aims to achieve the invariant across source domains. We experiment with `G2DM` (Albuquerque et al., 2019), `DANN` (Ganin et al., 2016), `CDANN` (Li et al., 2018c), `CORAL` (Sun & Saenko, 2016), `IRM` (Arjovsky et al., 2019). These models focus on accuracy transfer by enforcing the invariance of distributions $P_{D_i^S}^{Z}$ or $P_{D_i^S}^{Z|Y}$.
- Distributionally robust optimization: Method that learns a model at worst-case distribution to hope it can generalize well on test data. We experiment with `GroupDRO` (Sagawa et al., 2019) that minimizes the worst-case training loss over a set of pre-defined groups through regularization.
- `ATDM`: A variant of `FATDM-StarGAN` that solely focuses on accuracy transfer. That is, we only enforce the invariance of $P_{D_i^S}^{Z|Y}$ during learning which is similar to Nguyen et al. (2021).

The implementations of these models except `G2DM` are adapted from DomainBed framework (Gulrajani & Lopez-Paz, 2020). For `G2DM`, we use the author-provided implementation. For all models, we use ResNet18 (He et al., 2016) as the backbone module of representation mapping $g : \mathcal{X} \rightarrow \mathcal{Z}$; and fairness constraint $\mathcal{L}_{fair}$ is enforced as a regularization term added to the original objective functions.

**Experiment setup.** We follow leave-one-out domain setting in which 3 domains are used for training and the remaining domain serves as the unseen target domain and is used for evaluation. Several metrics are considered to measure the unfairness and error of each model in target domain, including:

- Error: cross-entropy loss (CE), misclassification rate (MR), $\overline{\text{AUROC}} := 1 - \text{AUROC}$, $\overline{\text{AUPR}} := 1 - \text{AUPR}$, $\overline{F_1} := 1 - F_1$, where AUROC, AUPR, $F_1$ are area under receiver operating characteristic curve, area under precision-recall curve, $F_1$ score, respectively.
- Unfairness: we consider both *equalized odds* and *equal opportunity* fairness notion, and adopt mean distance (MD) and earth mover's distance (EMD) as distance metric $\mathcal{D}(\cdot || \cdot)$.

**Fairness and accuracy on target domains.** We first compare our method with baselines in terms of the optimal trade-off (Pareto frontier) between accuracy and fairness on target domains under different metric pairs. Figure 4 shows the error-unfairness curves (as $\omega$ varies from 0 (no fairness constraint) to 10 (strong fairness constraint)), with $\overline{\text{AUROC}}$ and MR as error metric, and equalized odds (measured under distance metrics MD and EMD) as fairness notion; the results for other error metrics are similar

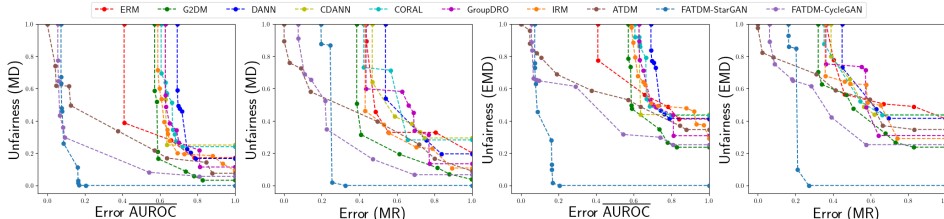

Figure 4: Fairness-accuracy trade-off (Pareto frontier) of `FATDM-StarGAN`, `FATDM-CycleGAN`, and baseline methods: error-unfairness curves are constructed by varying $\omega \in [0, 10]$ and the values of error and unfairness are normalized to $[0, 1]$. Lower-left points indicate the model has a better fairness-accuracy trade-off (Pareto optimality).

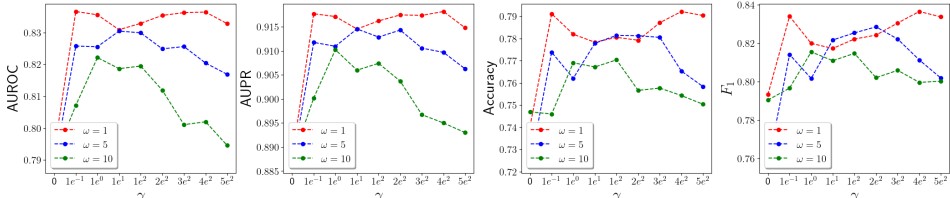

Figure 5: Prediction performances (AUROC, AUPR, Accuracy, $F_1$) of `FATDM-StarGAN` on Cardiomegaly disease data when varying hyper-parameter $\gamma$ at different levels of fairness constraint $\omega$.

and shown in Appendix C. Our observations are as follows: (1) As expected, there is a trade-off between fairness and accuracy: for all methods, increasing $\omega$ improves fairness but reduces accuracy. (2) Among all methods, the Pareto frontiers of `FATDM-StarGAN` and `FATDM-CycleGAN` are the bottom leftmost, implying that our method attains a better fairness-accuracy trade-off than baselines. (3) Although fairness constraint is imposed during training for all methods, the fairness attained at source domains cannot be well-generalized to the target domain under other methods. These results validate our theorems and show that enforcing the domain-invariant $P_{D_i^S}^{Z|Y}$ and $P_{D_i^S}^{Z|Y,A}$ when learning representations ensures the transfer of both accuracy and fairness. It is worth-noting that under this dataset, the domain-invariant $P_{D_i^S}^{Z|Y}$ (accuracy transfer) does not imply the domain-invariant $P_{D_i^S}^{Z|Y,A}$ (fairness transfer). This is because domain $D_i^S$ (i.e., age) is correlated with label $Y$ (i.e., has a disease) and sensitive attribute $A$ (i.e., gender), making the distribution $P_{D_i^S}^{Y,A}$ different across domains.

**Impact of density mapping model.** To investigate whether the performance gain of our method is due to the use of any specific density mapping model, we adopt `StarGAN` and `CycleGAN` architectures to learn density mapping functions in our method and compare their performances. Figure 4 shows that `FATDM-StarGAN` and `CycleGAN` achieve similar fairness-accuracy trade-off at the target domains and both of them outperform the baselines. This result shows that our method is not limited to any specific density mapping model and is broadly applicable to other architectures.

**Impact of invariant representation constraints.** We also examine the impact of $\mathcal{L}_{inv}$ on the performance of `FATDM-StarGAN` at target domains, where we vary the hyper-parameter $\gamma \in [0, 5e^2]$ at different levels of fairness (i.e., fix $\omega = 1, 5, 10$) and examine how the prediction performances (i.e., AUROC, AUPR, accuracy and $F_1$) could change. Figure 5 shows that enforcing domain-invariant constraint $\mathcal{L}_{inv}$ helps transfer the performance from source to target domain, and $\gamma$ that attains the highest accuracy at target domain can be different for different levels of fairness. The results also indicate the fairness-accuracy trade-off, i.e., for any $\gamma$, enforcing stronger fairness constraints (large $\omega$) could hurt prediction performances.

## 6 CONCLUSION

In this paper, we theoretically and empirically demonstrate how to achieve fair and accurate predictions in unknown testing environments. To the best of our knowledge, our work provides the first theoretical analysis to understand the efficiency of invariant representation learning in transferring both fairness and accuracy under domain generalization. In particular, we first propose the upper bounds of prediction error and unfairness in terms of JS-distance, then design the two-stage learning method that minimizes these upper bounds by learning domain-invariant representations. Experiments on the real-world clinical data demonstrate the effectiveness of our study.

## REPRODUCIBILITY STATEMENT

The original chest X-ray images and the corresponding metadata can be downloaded from PhysioNet (`https://physionet.org/content/mimic-cxr-jpg/2.0.0/`; `https://physionet.org/content/mimiciv/2.0/`). Codes for data processing and proposed algorithms are in supplementary materials. Technical details of the proposed algorithms and experimental settings are in Appendix B. Additional experimental results are in Appendix C. Lemmas used in proofs of the theorems in the main paper are in Appendix D. Complete proofs of the theorems in the main paper and the corresponding lemmas are in Appendix E.

## ACKNOWLEDGEMENTS

This work was funded in part by the National Science Foundation under award number IIS-2145625, by the National Institutes of Health under award number UL1TR002733, and by The Ohio State University President's Research Excellence Accelerator Grant.

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

## A    RELATED WORKS

**Domain generalization/Domain adaptation:** In many real scenarios of machine learning, data in training phase is sampled from one or many source domains, while in the testing phase, data is sampled from an unseen target domain. Many works have been proposed to design robust ML models that can achieve good performances in deployment environment depending on whether they can access to the target data (domain adaptation) or not (domain generalization). However, most of these models focus only on transfering accuracy from source to target domains and can be categorized into five main approaches: (1) data manipulation (Volpi et al., 2018; Qiao et al., 2020; Zhou et al., 2020; Zhang et al., 2018; Shankar et al., 2018); (2) domain-invariant representation learning (Li et al., 2018b;a; Ganin & Lempitsky, 2015; Ganin et al., 2016; Phung et al., 2021; Nguyen et al., 2021); (3) distributional robustness (Krueger et al., 2021; Liu et al., 2021; Koh et al., 2021; Wang et al., 2021; Sagawa et al., 2019; Hu et al., 2018), (4) gradient operation (Huang et al., 2020; Shi et al., 2021; Rame et al., 2021; Tian et al., 2022), and (5) self-supervised learning (Carlucci et al., 2019; Kim et al., 2021; Jeon et al., 2021; Li et al., 2021).

**Fairness in Machine Learning:** Many fairness notions have been proposed to measure the unfairness in ML model, and they can be roughly classified into two classes: *Individual fairness* considers the equity at the individual-level and it requires that similar individuals should be treated similarly (Biega et al., 2018; Bechavod et al., 2020; Gupta & Kamble, 2021; Dwork et al., 2012). *Group fairness* attains a certain balance in the group-level, where the entire population is first partitioned into multiple groups and certain statistical measures are equalized across different groups (Hardt et al., 2016; Zhang et al., 2019; 2020). Various approaches have also been developed to satisfy these fairness notions, they roughly fall into three categories: (1) *Pre-processing*: modifying training dataset to remove bias before learning an ML model (Kamiran & Calders, 2012; Zemel et al., 2013). (2) *In-processing*: attain fairness during the training process by imposing certain fairness constraint or modifying loss function. (Zafar et al., 2019; Agarwal et al., 2018) (3) *Post-processing*: altering the output of an existing algorithm to satisfy a fairness constraint after training (Hardt et al., 2016). However, most of these methods assume the data distributions at training and testing are the same. In contrast, we study fairness problem under domain generalization in this paper.

**Fairness under Domain Adaptation:** There are some studies proposed to achieve good fairness when the testing environment changes but all of them focused on the domain adaptation setting. The most common adaptation setup is learning under the assumption of covariate shift. For example, Singh et al. (2021) leveraged a feature selection method in a causal graph describing data to mitigate fairness violation under covariate shift of distribution in testing data. Coston et al. (2019) proposed the weighting methods that can give fair prediction under covariate shift between source and target distribution when access to the sensitive attributes is prohibited. Rezaei et al. (2021) sought fair decisions by optimizing a worst-case testing performance. Besides convariate shift, there are some works proposed to handle other types of distribution shift including demographic shift and prior probability shift. Instead of learning fair model directly, Oneto et al. (2019) and Madras et al. (2018) find fair representation that can generalize to the new tasks. Aside from empirical studies, Schumann et al. (2019) and Yoon et al. (2020) developed theoretical frameworks to examine fairness transfer in domain adaptation setting and then offered modeling approaches to achieve good fairness in the target domain.

**Comparison with existing bounds in the literature:** We compare our bounds with most commons bound in the fields of domain adaptation and domain generalization as follows.

*Accuracy bounds in domain adaptation.*

- Bounds in Ben-David et al. (2010):

$$\epsilon_{D^T}^{\text{Acc}}\left(\widehat{f}\right) \leq \epsilon_{D^S}^{\text{Acc}}\left(\widehat{f}\right) + \mathcal{D}_{TV}\left(P_{D^S}^X \parallel P_{D^T}^X\right) + \min_{D \in \{D^S, D^T\}} \mathbb{E}_D\left[|f_{D^S}(X) - f_{D^T}(X)|\right]$$

This bound is for binary classification problem under domain adaptation. The classification error in target domain is bounded by the error in source domain, the total variation distance of feature distribution between source and target domain, and the misalignment of the labeling function between source and target domain. The limitation of this bound is that (1) it's only applicable to settings with zero-one loss function and deterministic labeling function; (2) estimating the total variation distance is hard in practice and it doesn't relate the feature and representation spaces.

This paper also provides another accuracy bound based on $\mathcal{H}\Delta\mathcal{H}$ divergence:.

$$\epsilon_{D^T}^{\text{Acc}}\left(\widehat{f}\right) \le \epsilon_{D^S}^{\text{Acc}}\left(\widehat{f}\right) + \mathcal{D}_{\mathcal{H}\Delta\mathcal{H}}\left(P_{D^S}^X \parallel P_{D^T}^X\right) + \inf_{\widehat{f}}\left[\epsilon_{D^T}^{\text{Acc}}\left(\widehat{f}\right) + \epsilon_{D^S}^{\text{Acc}}\left(\widehat{f}\right)\right]$$

where $\mathcal{D}_{\mathcal{H}\Delta\mathcal{H}}\left(P_{D^S}^X \parallel P_{D^T}^X\right) = \sup_{\widehat{f}_1,\widehat{f}_1}\left|P_{D^S}\left(\widehat{f}_1(X) \ne \widehat{f}_2(X)\right) - P_{D^T}\left(\widehat{f}_1(X) \ne \widehat{f}_2(X)\right)\right|$ is the

$\mathcal{H}\Delta\mathcal{H}$ divergence. However, it has the same limitations as total variation distance mentioned above.

*Accuracy bounds in domain generalization.*

- Bounds in Albuquerque et al. (2019):

$$\epsilon_{D^T}^{\text{Acc}}\left(\widehat{f}\right) \le \sum_{i=1}^{N}\pi_i\epsilon_{D_i^S}^{\text{Acc}}\left(\widehat{f}\right) + \max_{j,k\in[N]}\mathcal{D}_{\mathcal{H}}\left(P_{D_j^S}^X \parallel P_{D_k^S}^X\right) + \mathcal{D}_{\mathcal{H}}\left(P_{D_*^S}^X \parallel P_{D^T}^X\right)$$
$$+ \min_{D\in\{D_*^S,D^T\}}\mathbb{E}_D\left[\left|f_{D_*^S}(X) - f_{D^T}(X)\right|\right]$$

where $\mathcal{D}_{\mathcal{H}}\left(P_{D^S}^X \parallel P_{D^T}^X\right) = \sup_{\widehat{f}}\left|P_{D^S}\left(\widehat{f}(X) = 1\right) - P_{D^T}\left(\widehat{f}(X) = 1\right)\right|$ is the $\mathcal{H}$ divergence,

$P_{D_*^S}^X = \arg\min_{\pi}\mathcal{D}_{\mathcal{H}}\left(\sum_{i=1}^{N}\pi_i P_{D_i^S}^X \parallel P_{D^T}^X\right)$ is the mixture of source domains that is closest to

target domain with respect to $\mathcal{H}$ divergence. In this bound, the classification error in target domain is bounded by the convex combination of errors in source domains, the $\mathcal{H}$ divergence between source domains, the $\mathcal{H}$ divergence between target domain and its nearest mixture of source domains, and the misalignment of the labeling function between mixture source domains and target domain. Because this bound is constructed based on $\mathcal{H}$ divergence, it also has the limitations for the bounds in domain adaptation (Ben-David et al., 2010) as we mentioned. This bound can be transformed to the representation space $\mathcal{Z}$ by replacing $X$ by $Z$ in its formula. Then, this bound suggests enforcing invariant constraint of marginal distribution of representation $Z$ across source domains, which has inherent trade-off as shown in Thm. 2. Because the target domain is unknown during training, the mixing weights $\{\pi_i\}_{i=1}^{N}$ are not useful for algorithmic design.

- Bounds in Phung et al. (2021):

$$\epsilon_{D^T}^{\text{Acc}}\left(\widehat{f}\right) \le \sum_{i=1}^{N}\pi_i\epsilon_{D_i^S}^{\text{Acc}}\left(\widehat{f}\right) + C\max_{i\in[N]}\mathbb{E}_{D_i^S}\left[\left\|\left[\left|f_{D^T}(X)_y - f_{D_i^S}(X)_y\right|\right]_{y=1}^{|\mathcal{Y}|}\right\|_1\right]$$
$$+ \sum_{i=1}^{N}\sum_{j=1}^{N}\frac{C\sqrt{2\pi_j}}{N}d_{1/2}\left(P_{D^T}^Z, P_{D_i^S}^Z\right) + \sum_{i=1}^{N}\sum_{j=1}^{N}\frac{C\sqrt{2\pi_j}}{N}d_{1/2}\left(P_{D_i^S}^Z, P_{D_j^S}^Z\right)$$

where $d_{1/2}\left(P_{D_i^S}^X, P_{D_j^S}^X\right) = \sqrt{\mathcal{D}_{1/2}\left(P_{D_i^S}^X \parallel P_{D_j^S}^X\right)}$ is Hellinger distance defined based on

Hellinger divergence $\mathcal{D}_{1/2}\left(P_{D_i^S}^X \parallel P_{D_j^S}^X\right) = 2\int_{\mathcal{X}}\left(\sqrt{P_{D_i^S}^X} - \sqrt{P_{D_j^S}^X}\right)^2 dX$. This bound relates the feature and representation spaces that the classification error of target domain defined in feature space is bounded by classification errors of source domains defined in feature space, the misalignment of labeling function between target and source domains, and the Hellinger distances between source and target domains and between source domains of marginal distribution of representation $Z$. While this bound is not limited to zero-one loss and the labeling function can be stochastic, it suggests the alignment of marginal distribution of representation $Z$ across source domains for generalization. Moreover, estimating Hellinger distance can be hard in practice.

*The mismatch between existing bounds and adversarial learning approach for domain generalization.*

All existing bounds mentioned above suggest minimizing the distances between representation distributions across source domains with respect to some discrepancy measures such as $\mathcal{H}$ divergence, total variation distance, and Hellinger distance. Based on these bounds, adversarial learning-based models are often proposed to minimize these distances. However, there is a misalignment between the objectives of adversarial learning and the bounds which results in the gap between theoretical findings and practical algorithms.

In particular, it has been shown that the objective of the minimax game between the representation mapping and the discriminator is equivalent to minimizing the JS divergence between representation distributions across source domains (Goodfellow et al., 2014). However, minimizing JS divergence does not guarantee the minimization of common distances used in the existing bounds. The details are as follows.

- $\mathcal{H}$ divergence: We show that JS divergence is not the upper bound of $\mathcal{H}$ divergence. Consider an example with two distributions $P(X)$ and $Q(X)$ where $\begin{cases} P(X) = 0 & w.p \ 1/3 \\ P(X) = 1 & w.p \ 2/3 \end{cases}$ and $\begin{cases} Q(X) = 0 & w.p \ 1/3 \\ Q(X) = 1 & w.p \ 2/3 \end{cases}$. By definition, $\mathcal{D}_{\mathcal{H}}(P \parallel Q) \sim 0.33 > \mathcal{D}_{JS}(P \parallel Q) \sim 0.08$.

- Total variation distance: We have $\mathcal{D}_{JS}(P \parallel Q) \le \mathcal{D}_{TV}(P \parallel Q) \ \forall P, Q$ where $\mathcal{D}_{JS}$ and $\mathcal{D}_{TV}$ are JS divergence and total variation distance, respectively. Then, minimizing JS divergence does not guarantee the minimization of total variation distance.

- Hellinger distance: We have $\mathcal{D}_{JS}(P \parallel Q) \le \sqrt{2} d_{1/2}(P, Q) \ \forall P, Q$ where $d_{1/2}$ is Hellinger distance and total variation distance, respectively. Then, minimizing JS divergence does not guarantee the minimization of Hellinger distance.

Different from the existing bounds, our bounds are based on JS divergence/distance. Then they align with the adversarial learning approach for domain generalization in general, and with our proposed method FATDM in particular.

*Advantages of our proposed bounds in domain generalization.*

In summary, our proposed bounds has several advantages in terms of the following:

- Most existing bounds (Ben-David et al., 2010; Albuquerque et al., 2019) do not relates feature and representation spaces so it is not clear how performance in input space is affected by the representations. In contrast, our bounds connect the representation and input spaces; this further guides us to find representations that lead to good performances in input space.

- Most prior studies adopt $\mathcal{H}$ divergence to measure the dissimilarity between domains, which is limited to deterministic labeling functions and zero-one loss (Ben-David et al., 2010; Albuquerque et al., 2019). In contrast, our bound is more general and is applicable to settings where domains are specified by stochastic labeling functions and general loss functions.

- Distant metrics (i.e., total variation distance, $\mathcal{H}$ divergence, Hellinger divergence, etc.) used in existing bounds (Ben-David et al., 2010; Albuquerque et al., 2019; Phung et al., 2021) are hard to compute in practice. In contrast, our bounds use JS divergence which is aligned with training objective for discriminator in adversarial learning Goodfellow et al. (2014).

- Existing bounds for domain generalization only imply the alignment of marginal distribution of feature across source domains (Albuquerque et al., 2019; Phung et al., 2021). As shown in Thm. 2, methods that learn invariance of marginal distribution have an inherent trade-off and may increase the lower bound of expected loss. In contrast, our bounds suggest the alignment of label-conditional distribution of feature across source domains which has been verified to be more effective in empirical studies (Li et al., 2018b;c; Zhao et al., 2020; Nguyen et al., 2021).

- Regarding the fairness, our work is the first that bounds the unfairness in domain generalization. In particular, our bounds suggest enforcing the invariant constraint of feature distribution given label and sensitive attribute across source domains to transfer fairness to the unseen target domain.

## B DETAILS OF ALGORITHM FATDM

FATDM consists of density mapping functions $m_{i,j}^y$ and $m_{i,j}^{y,a}$, $\forall y \in \mathcal{Y}, a \in \mathcal{A}, i, j \in [N]$ (learned by two DensityMatch models), feature mapping function $g$ (ResNet18 model), and the classifier $\hat{h}$. In our study, we experiment with two different DensityMatch architectures: StarGAN (i.e., in FATDM-StarGAN) and CycleGAN (in FATDM-CycleGAN). We show the details of FATDM-StarGAN below. For FATDM-CycleGAN, the only difference is we used CycleGAN as DensityMatch instead of StarGAN. The details of CycleGAN were presented in the original paper (Zhu et al., 2017).

For `FATDM-StarGAN`, each `DensityMatch`$^Y$ (or `DensityMatch`$^{Y,A}$) consists of a *generator* $\mathsf{G} : \mathcal{X} \times [N] \times [N] \to \mathcal{X}$ and a *discriminator* $\mathsf{D} : \mathcal{X} \to [N] \times \{0, 1\}$. The generator takes in real image $x$ and a pair of domain labels $i, j$ as input and generates a fake image; the discriminator aims to predict the domain label of the image generated by the generator and distinguish whether it is fake or real. $\mathsf{G}$ and $\mathsf{D}$ are learned simultaneously by solving the following optimizations:

$$\textbf{Discriminator's objective:} \quad \min \mathcal{L}_{\mathsf{D}}^{\texttt{StarGAN}} := -\mathcal{L}_{adv}^{\texttt{StarGAN}} + \lambda_{cls}\mathcal{L}_{cls(\text{real})}^{\texttt{StarGAN}}$$

$$\textbf{Generator's objective:} \quad \min \mathcal{L}_{\mathsf{G}}^{\texttt{StarGAN}} := \mathcal{L}_{adv}^{\texttt{StarGAN}} + \lambda_{cls}\mathcal{L}_{cls(\text{fake})}^{\texttt{StarGAN}} + \lambda_{rec}\mathcal{L}_{rec}^{\texttt{StarGAN}} \qquad (6)$$

where $\mathcal{L}_{adv}^{\texttt{StarGAN}}$ is the adversarial loss, $\mathcal{L}_{cls(\text{fake})}^{\texttt{StarGAN}}, \mathcal{L}_{cls(\text{real})}^{\texttt{StarGAN}}$ are domain classification loss with respect to fake and real images respectively, $\mathcal{L}_{rec}^{\texttt{StarGAN}}$ is the reconstruction loss. The specific formulations of these loss functions are in Choi et al. (2018). $\lambda_{cls}$ and $\lambda_{rec}$ are hyper-parameters that control the relative importance of domain classification and reconstruction losses, respectively, compared to the adversarial loss.

In our experiments, input images are resized to $(256, 256)$ and normalized into the range $[-1, 1]$. The dimension of representation space $\mathcal{Z}$ is set to $512$. $\omega$ (hyper-parameter that controls accuracy-fairness trade-off) varies from 0 to 10 with step sizes $0.0002$ for $\omega \in [0, 0.002]$, $0.002$ for $\omega \in [0.002, 0.1]$ and $0.2$ for $\omega \in [0.2, 10]$, and $\gamma$ (hyper-parameter that controls accuracy-invariance trade-off) is set to $0.1$ (after hyper-parameter tuning). Models (`FATDM` and baselines) are implemented by PyTorch library version 1.11 and is trained on multiple computer nodes (each model instance is trained on a single node which has 4 CPUs, 8GB of memory, and a single GPU (P100 or V100)). One domain's data is used for testing and the other domains' data is used for training ($10\%$ of training data is used for validation). Each model is trained with 10 epochs and the results are from the epoch with best performance on the validation set. Figure 6 visualizes the two-stage training process of `FATDM-StarGAN`. The detailed architectures of `FATDM-StarGAN` are shown in Tables 2-5. We have also provided all code for these models in supplemental material.

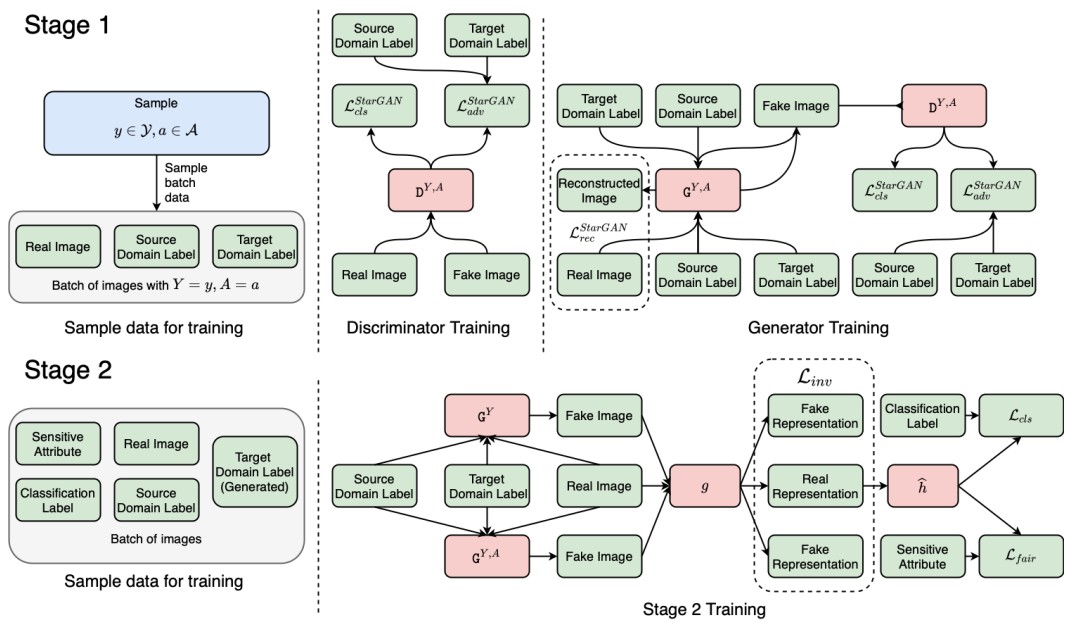

Figure 6: Two-stage training of `FATDM-StarGAN`. For stage 1, we only show the training process for `DensityMatch`$^{Y,A}$ (training process for `DensityMatch`$^Y$ is similar.)

Table 2: Architecture of StarGAN generators $\mathsf{G}^Y$ and $\mathsf{G}^{Y,A}$ - Density mapping functions $m_{i,j}^y$ and $m_{i,j}^{y,a} \; \forall y \in \mathcal{Y}, a \in \mathcal{A}, i, j \in [N]$. This architecture is similar to the one in the original paper Choi et al. (2018) except for the first convolution layer where number of input channels is 1 (for grayscale images) and input shape is $(h, w, 1 + 2n_c)$. $(h, w)$ is the size of input images, IN is instance batchnorm, and ReLU is Rectified Linear Unit. N: number of output channels, K: kernel size, S: stride szie, P: padding size are convolution and deconvolution layers' hyper-parameters.

| Part | Input → Output Shape | Layer Information |
|---|---|---|
| Down-sampling | $(h, w, 1 + 2n_c) \rightarrow (h, w, 64)$ | CONV-(N64, K7x7, S1, P3), IN, ReLU |
| | $(h, w, 64) \rightarrow \left(\frac{h}{2}, \frac{w}{2}, 128\right)$ | CONV-(N128, K4x4, S2, P1), IN, ReLU |
| | $\left(\frac{h}{2}, \frac{w}{2}, 128\right) \rightarrow \left(\frac{h}{4}, \frac{w}{4}, 256\right)$ | CONV-(N256, K4x4, S2, P1), IN, ReLU |
| Bottleneck | $\left(\frac{h}{4}, \frac{w}{4}, 256\right) \rightarrow \left(\frac{h}{4}, \frac{w}{4}, 256\right)$ | Residual Block: CONV-(N256, K3x3, S1, P1), IN, ReLU |
| | $\left(\frac{h}{4}, \frac{w}{4}, 256\right) \rightarrow \left(\frac{h}{4}, \frac{w}{4}, 256\right)$ | Residual Block: CONV-(N256, K3x3, S1, P1), IN, ReLU |
| | $\left(\frac{h}{4}, \frac{w}{4}, 256\right) \rightarrow \left(\frac{h}{4}, \frac{w}{4}, 256\right)$ | Residual Block: CONV-(N256, K3x3, S1, P1), IN, ReLU |
| | $\left(\frac{h}{4}, \frac{w}{4}, 256\right) \rightarrow \left(\frac{h}{4}, \frac{w}{4}, 256\right)$ | Residual Block: CONV-(N256, K3x3, S1, P1), IN, ReLU |
| | $\left(\frac{h}{4}, \frac{w}{4}, 256\right) \rightarrow \left(\frac{h}{4}, \frac{w}{4}, 256\right)$ | Residual Block: CONV-(N256, K3x3, S1, P1), IN, ReLU |
| | $\left(\frac{h}{4}, \frac{w}{4}, 256\right) \rightarrow \left(\frac{h}{4}, \frac{w}{4}, 256\right)$ | Residual Block: CONV-(N256, K3x3, S1, P1), IN, ReLU |
| Up-sampling | $\left(\frac{h}{4}, \frac{w}{4}, 256\right) \rightarrow \left(\frac{h}{2}, \frac{w}{2}, 128\right)$ | DECONV-(N128, K4x4, S2, P1), IN, ReLU |
| | $\left(\frac{h}{2}, \frac{w}{2}, 128\right) \rightarrow (h, w, 64)$ | DECONV-(N64, K4x4, S2, P1), IN, ReLU |
| | $(h, w, 64) \rightarrow (h, w, 3)$ | CONV-(N3, K7x7, S1, P3), IN, ReLU |

Table 3: Architecture of StarGAN discriminators. This architecture is similar to the one in the original paper Choi et al. (2018) except for the first convolution layer where number of input channels is 1 (for grayscale images). $(h, w)$ is the size of input images, $n_d$ is the number of domains, and Leaky ReLU is Leaky Rectified Linear Unit. N: number of output channels, K: kernel size, S: stride szie, P: padding size are convolution layers' hyper-parameters.

| Layer | Input $\to$ Output Shape | Layer Information |
|---|---|---|
| Input Layer | $(h, w, 1) \to \left(\frac{h}{2}, \frac{w}{2}, 64\right)$ | CONV-(N64, K4x4, S2, P1), Leaky ReLU |
| Hidden Layer | $\left(\frac{h}{2}, \frac{w}{2}, 64\right) \to \left(\frac{h}{4}, \frac{w}{4}, 128\right)$ | CONV-(N128, K4x4, S2, P1), Leaky ReLU |
| Hidden Layer | $\left(\frac{h}{4}, \frac{w}{4}, 128\right) \to \left(\frac{h}{8}, \frac{w}{8}, 256\right)$ | CONV-(N256, K4x4, S2, P1), Leaky ReLU |
| Hidden Layer | $\left(\frac{h}{8}, \frac{w}{8}, 256\right) \to \left(\frac{h}{16}, \frac{w}{16}, 512\right)$ | CONV-(N512, K4x4, S2, P1), Leaky ReLU |
| Hidden Layer | $\left(\frac{h}{16}, \frac{w}{16}, 512\right) \to \left(\frac{h}{32}, \frac{w}{32}, 1024\right)$ | CONV-(N1024, K4x4, S2, P1), Leaky ReLU |
| Hidden Layer | $\left(\frac{h}{32}, \frac{w}{32}, 1024\right) \to \left(\frac{h}{64}, \frac{w}{64}, 2048\right)$ | CONV-(N2048, K4x4, S2, P1), Leaky ReLU |
| Output Layer ($D_{src}$) | $\left(\frac{h}{64}, \frac{w}{64}, 2048\right) \to \left(\frac{h}{64}, \frac{w}{64}, 1\right)$ | CONV-(N1, K3x3, S1, P1) |
| Output Layer ($D_{cls}$) | $\left(\frac{h}{64}, \frac{w}{64}, 2048\right) \to (1, 1, n_d)$ | CONV-(N($n_d$), K$\frac{h}{64} \times \frac{w}{64}$, S1, P0) |

Table 4: Architecture of feature mapping $g$. This architecture is similar to ResNet18 model He et al. (2016) except for the first convolution layer where number of input channels is 1 (for grayscale images) and the last layer where output dimension is $n_z$ - dimension of representation space $\mathcal{Z}$. $(h, w)$ is the size of input images, BN is batchnorm, MaxPool is max pooling, AvePool is average pooling, and ReLU is Rectified Linear Unit. N: number of output channels, K: kernel size, S: stride szie, P: padding size are convolution layers' hyper-parameters.

| Part | Input $\to$ Output Shape | Layer Information |
|---|---|---|
| Input | $(h, w, 1) \to \left(\frac{h}{2}, \frac{w}{2}, 64\right)$ | CONV-(N64, K7x7, S2, P3), BN, ReLU, MaxPool |
| Bottleneck | $\left(\frac{h}{2}, \frac{w}{2}, 64\right) \to \left(\frac{h}{4}, \frac{w}{4}, 64\right)$ | Residual Block: CONV-(N64, K3x3, S1, P1), BN, ReLU, CONV-(N64, K3x3, S1, P1), BN |
| | $\left(\frac{h}{4}, \frac{w}{4}, 64\right) \to \left(\frac{h}{8}, \frac{w}{8}, 128\right)$ | Residual Block: CONV-(N128, K3x3, S1, P1), BN, ReLU, CONV-(N128, K3x3, S1, P1), BN |
| | $\left(\frac{h}{8}, \frac{w}{8}, 128\right) \to \left(\frac{h}{16}, \frac{w}{16}, 256\right)$ | Residual Block: CONV-(N256, K3x3, S1, P1), BN, ReLU, CONV-(N256, K3x3, S1, P1), BN |
| | $\left(\frac{h}{16}, \frac{w}{16}, 256\right) \to (1, 1, 512)$ | Residual Block: CONV-(N512, K3x3, S1, P1), IN, ReLU, CONV-(N512, K3x3, S1, P1), BN, AvgPool |
| Output | $(1, 1, 512) \to n_z$ | LINEAR-(512, $n_z$) |

Table 5: Architecture of classifier $\widehat{h}$. $n_z$ is the dimension of representation space $\mathcal{Z}$.

| Layer | Input $\to$ Output Shape | Layer Information |
|---|---|---|
| Hidden Layer | $n_z \to \frac{n_z}{2}$ | LINEAR-$\left(n_z, \frac{n_z}{2}\right)$, ReLU |
| Hidden Layer | $\frac{n_z}{2} \to \frac{n_z}{4}$ | LINEAR-$\left(\frac{n_z}{2}, \frac{n_z}{4}\right)$, ReLU |
| Output Layer | $\frac{n_z}{4} \to 1$ | LINEAR-$\left(\frac{n_z}{4}, 1\right)$, Sigmoid |

## C    ADDITIONAL EXPERIMENTS

**Experimental results with all unfairness and error metrics.**    In this section, we provide more experimental results about fairness and accuracy under domain generalization. In particular, we investigate fairness-accuracy trade-off on the two clinical image datasets including Cardiomegaly and Edema diseases with respect to different fairness criteria (i.e., Equalized Odds, Equal Opportunity), and unfairness (i.e., MD and EMD) and error (i.e., CE, MR, $\overline{AUROC}$, $\overline{AUPR}$, $\overline{F_1}$) measures. Figure 7 (Cardiomegaly disease - Equalized Odds), Figure 8 (Cardiomegaly disease - Equal Opportunity), Figure 9 (Edema disease - Equalized Odds), and Figure 10 (Edema disease - Equal Opportunity) show the unfairness-error curves of our models as well as baselines for these two datasets. As we can see, our model outperforms other baselines in terms of fairness-accuracy trade-off. The curve of our model is the bottom-leftmost compared to other baselines in all measures showing the clear benefit of (1) enforcing conditional invariant constraints for accuracy and fairness transfer and (2) using the two-stage training process to stabilize training compared to adversarial learning approach. We also quantify our observations by calculating the areas under these unfairness-error curves, in which the smaller area indicates the better accuracy-fairness trade-off. As shown in Tables 6 and 7, our model has the smallest areas under the curve and achieves significantly better fairness-accuracy trade-off for both equalized odd and equal opportunity compared to other methods.

**Impact of the number of source domains.**    Our work focuses on transferring fairness and accuracy under domain generalization when the target domain data are inaccessible during training. Instead, it relies on a set of source domains to generalize to an unseen, novel target domain. We investigate the relationship between the fairness-accuracy trade-off on the target domain and the number of source domains during training. In particular, we evaluate the performances of FATDM and ERM on Edema dataset with different numbers of source domains. Similar to the previous experiment, we first construct the dataset for each domain by rotating images with $\theta$ degree, where $\theta \in \{0°, 15°, 30°\}$ when the number of domain is 3, $\theta \in \{0°, 15°, 30°, 45°\}$ when the number of domain is 4, and $\theta \in \{0°, 15°, 30°, 45°, 60°\}$ when the number of domain is 5. The number of images per domain is adapted to ensure the training set size is fixed for the three cases. We follow the leave-one-out domain setting in which one domain serves as the unseen target domain for evaluation while the rest domains are for training; the average results across target domains are reported.

Figure 11 shows error-unfairness curves of FATDM and ERM when training with 2, 3, and 4 source domains. We observe that training with more source domains does not always help the model achieve better fairness-accuracy trade-off on unseen target domains. In particular, the performances of both FATDM and ERM are the best when training with 2 source domains and the worst when training with 3 source domains. We conjecture the reason that adding more source domains may help reduce the discrepancy between source and target domains (term (ii) in Thm. 1 and Thm. 3), but it may make it more difficult to minimize the source error and unfairness (term (i) in Thm. 1 and Thm. 3) and to learn invariant representation across the source domains (term (iii) in Thm. 1 and Thm. 3). Thus, our suggestion in practice is to conduct an ablation study to find the optimal number of source domains.

**Simultaneous and sequential training comparison.**    In all experiments we conducted so far, the fairness constraint $\mathcal{L}_{fair}$ is optimized simultaneously with the prediction error $\mathcal{L}_{acc}$ and the domain-invariant constraint $\mathcal{L}_{inv}$ for all methods. To investigate whether FATDM still attains a better accuracy-fairness trade-off when the processes of invariant representation learning and fair model training are decoupled, we conduct another set of experiments where models (FATDM (i.e., FATDM-StarGAN) and baselines G2DM, DANN, CDANN) are learned in a sequential matter: for each model, we first learn the representation mapping $g$ by optimizing $\mathcal{L}_{inv}$ and $\mathcal{L}_{acc}$; using the representations generated by the fixed $g$, we then learn the fair classifier by optimizing $\mathcal{L}_{acc}$ and $\mathcal{L}_{fair}$. The models trained based on the above procedure are named FATDM-seq, G2DM-seq, DANN-seq, and CDANN-seq; and their corresponding error-unfairness curves are shown in Figure 12. The results show that FATDM-seq still attains the best accuracy-fairness trade-off at target domain compared to G2DM-seq, DANN-seq, CDANN-seq. Our method is effective no matter whether $\mathcal{L}_{fair}$ and $\mathcal{L}_{inv}$ are optimized simultaneously or sequentially.

The reason that our method consistently outperforms the baselines for both settings is that the invariant-representation learning in baseline methods only guarantees the transfer of accuracy but not fairness. Even though a fairness regularizer is imposed to ensure the model is fair at source

domains (no matter whether invariant representations and fair classifier are trained simultaneously or sequentially), this fairness cannot be preserved at the target domain due to the potential distributional shifts. The key to ensuring the transfer of fairness is to learn representations such that $P(Z|Y, A)$ is domain-invariant; this must be done during the representation learning process. From Thm 3, we can see that unfairness at target domain $\epsilon_{D^T}^{\text{EO}}$ can still blow up if $P^{Z|Y,A}$ is different across domains, regardless of how fair the model is at source domains (i.e., small $\epsilon_{D_i^S}^{\text{EO}}$).

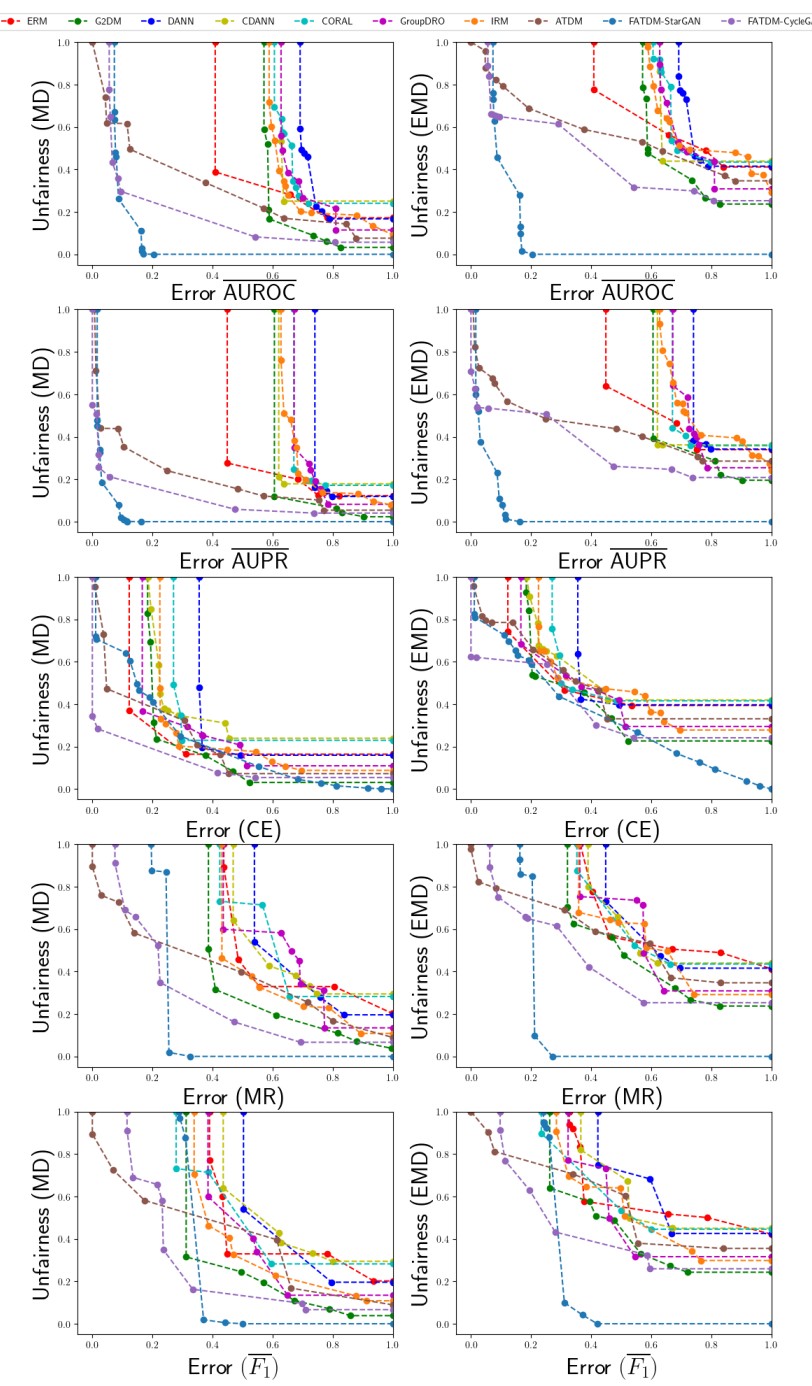

Figure 7: Error-unfairness curves with respect to equalized odds of FATDM and baselines on Cardiomegaly disease dataset.

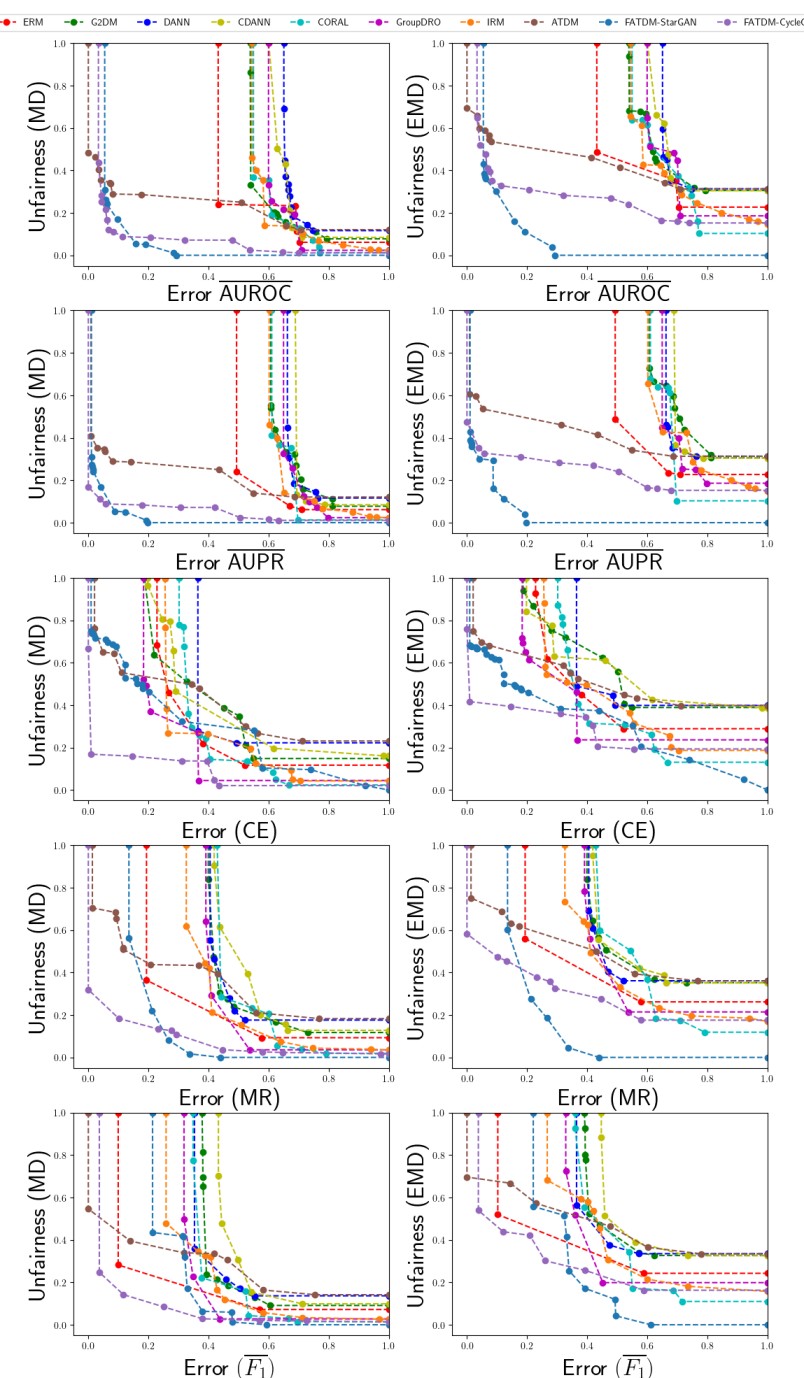

Figure 8: Error-unfairness curves with respect to equal opportunity of FATDM and baselines on Cardiomegaly disease dataset.

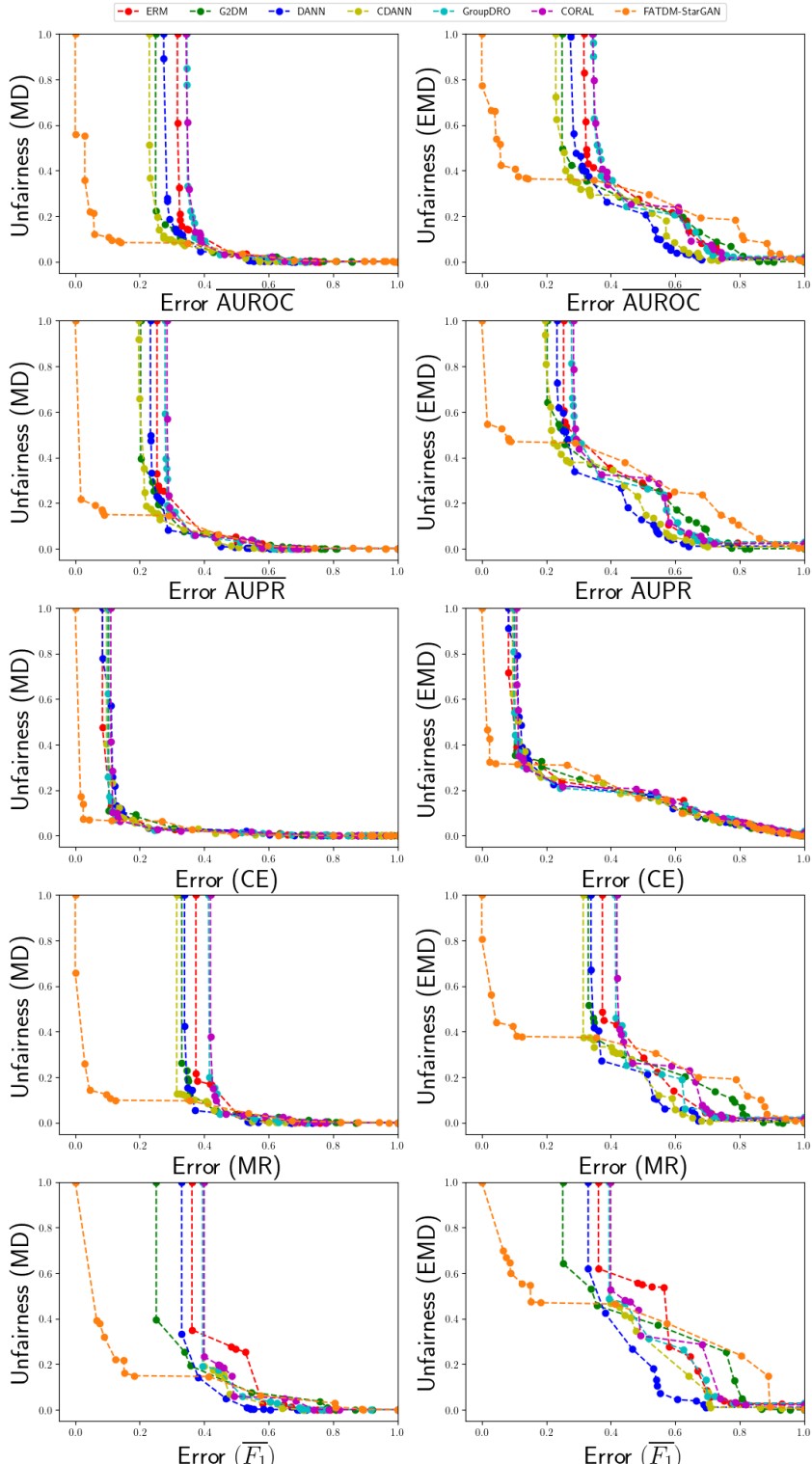

Figure 9: Error-unfairness curves with respect to equalized odds of FATDM and baselines on Edema disease dataset.

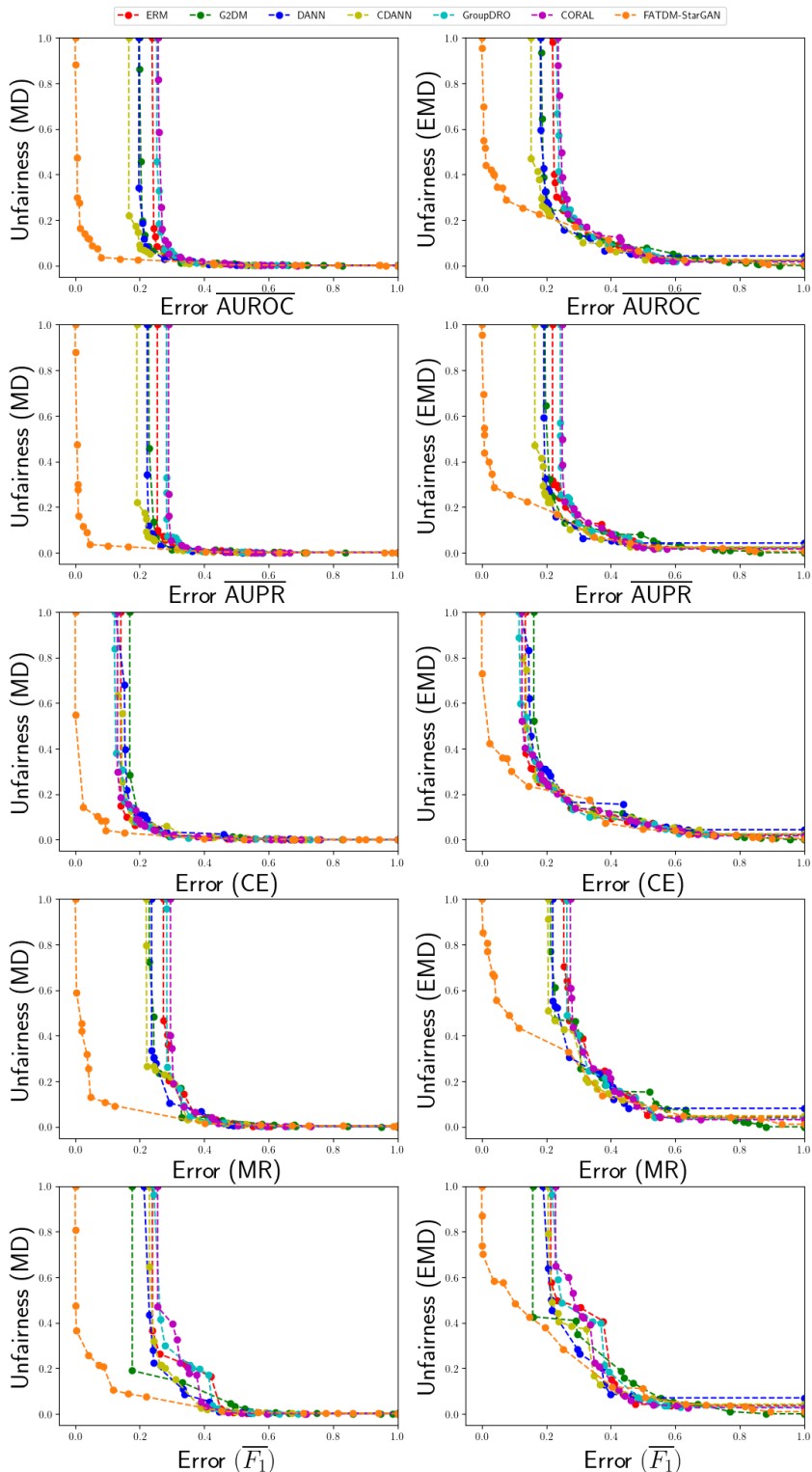

Figure 10: Error-unfairness curves with respect to equal opportunity of FATDM and baselines on Edema disease dataset.

Table 6: Area under the error-unfairness curves (Cardiomegaly disease dataset).

| Error - Unfairness | | Method | | | | | | | |
|---|---|---|---|---|---|---|---|---|---|
| | | ERM | G2DM | DANN | CDANN | CORAL | GroupDRO | IRM | FATDM |
| Equalized Odds | $\overline{AUROC}$ - MD | 0.5575 | 0.6093 | 0.7571 | 0.7224 | 0.7239 | 0.7039 | 0.6784 | **0.0935** |
| | $\overline{AUPRC}$ - MD | 0.5463 | 0.6301 | 0.7730 | 0.6883 | 0.7300 | 0.7152 | 0.6967 | **0.0291** |
| | CE - MD | 0.2861 | 0.2601 | 0.4622 | 0.4232 | 0.4424 | 0.3148 | 0.3370 | **0.2152** |
| | MR - MD | 0.6312 | 0.4906 | 0.6795 | 0.6667 | 0.6683 | 0.6382 | 0.5721 | **0.2439** |
| | $\overline{F_1}$ - MD | 0.5901 | 0.4150 | 0.6507 | 0.6547 | 0.5745 | 0.5360 | 0.5025 | **0.3365** |
| | $\overline{AUROC}$ - EMD | 0.7326 | 0.7106 | 0.8342 | 0.7931 | 0.8075 | 0.7845 | 0.7991 | **0.1099** |
| | $\overline{AUPRC}$ - EMD | 0.6901 | 0.7146 | 0.8308 | 0.7577 | 0.7918 | 0.7806 | 0.7945 | **0.0437** |
| | CE - EMD | 0.5158 | 0.4443 | 0.6143 | 0.5788 | 0.5873 | 0.4911 | 0.5274 | **0.3384** |
| | MR - EMD | 0.7056 | 0.5795 | 0.7137 | 0.6979 | 0.6902 | 0.6571 | 0.6483 | **0.2045** |
| | $\overline{F_1}$ - EMD | 0.6866 | 0.5328 | 0.7279 | 0.7019 | 0.6515 | 0.6027 | 0.6120 | **0.2888** |
| Equal Opportunity | $\overline{AUROC}$ - MD | 0.5128 | 0.6001 | 0.6999 | 0.6686 | 0.5935 | 0.6288 | 0.5910 | **0.0750** |
| | $\overline{AUPRC}$ - MD | 0.5419 | 0.6718 | 0.7086 | 0.7189 | 0.6423 | 0.6761 | 0.6435 | **0.0262** |
| | CE - MD | 0.3690 | 0.4272 | 0.5094 | 0.4492 | 0.3780 | 0.2737 | 0.3582 | **0.2754** |
| | MR - MD | 0.3203 | 0.5068 | 0.5252 | 0.5512 | 0.4897 | 0.4368 | 0.4173 | **0.1778** |
| | $\overline{F_1}$ - MD | 0.2134 | 0.4570 | 0.4608 | 0.5207 | 0.4017 | 0.3561 | 0.3510 | **0.2737** |
| | $\overline{AUROC}$ - EMD | 0.6119 | 0.7184 | 0.7649 | 0.7517 | 0.6720 | 0.7068 | 0.6780 | **0.0947** |
| | $\overline{AUPRC}$ - EMD | 0.6321 | 0.7684 | 0.7718 | 0.7877 | 0.6912 | 0.7335 | 0.7200 | **0.0448** |
| | CE - EMD | 0.5092 | 0.6093 | 0.6264 | 0.6141 | 0.4737 | 0.4340 | 0.4917 | **0.3070** |
| | MR - EMD | 0.4619 | 0.6420 | 0.6325 | 0.6532 | 0.5790 | 0.5515 | 0.5298 | **0.1918** |
| | $\overline{F_1}$ - EMD | 0.3876 | 0.6122 | 0.5942 | 0.6496 | 0.5101 | 0.4898 | 0.4889 | **0.3108** |

Table 7: Area under the error-unfairness curves (Edema disease dataset).

| Error - Unfairness | | Method | | | | | | |
|---|---|---|---|---|---|---|---|---|
| | | ERM | G2DM | DANN | CDANN | CORAL | GroupDRO | FATDM |
| Equalized Odds | $\overline{AUROC}$ - MD | 0.3395 | 0.2765 | 0.2972 | 0.2548 | 0.3642 | 0.3627 | **0.0633** |
| | $\overline{AUPRC}$ - MD | 0.2865 | 0.2446 | 0.2561 | 0.2304 | 0.3052 | 0.2998 | **0.0771** |
| | CE - MD | 0.1096 | 0.1266 | 0.1243 | 0.1192 | 0.1269 | 0.1179 | **0.0341** |
| | MR - MD | 0.3929 | 0.3525 | 0.3509 | 0.3302 | 0.4303 | 0.4240 | **0.0656** |
| | $\overline{F_1}$ - MD | 0.4213 | 0.3219 | 0.3527 | 0.4178 | 0.4283 | 0.4189 | **0.1369** |
| | $\overline{AUROC}$ - EMD | 0.4277 | 0.3813 | 0.3637 | 0.3419 | 0.4419 | 0.4394 | **0.2729** |
| | $\overline{AUPRC}$ - EMD | 0.3868 | 0.3588 | 0.3285 | 0.3245 | 0.3958 | 0.3921 | **0.3041** |
| | CE - EMD | 0.2366 | 0.2401 | 0.2348 | 0.2334 | 0.2447 | 0.2339 | **0.1827** |
| | MR - MD | 0.4592 | 0.4435 | 0.4017 | 0.3904 | 0.4942 | 0.4792 | **0.2802** |
| | $\overline{F_1}$ - MD | 0.5186 | 0.4642 | 0.4132 | 0.4827 | 0.5180 | 0.5029 | **0.3855** |
| Equal Opportunity | $\overline{AUROC}$ - MD | 0.2488 | 0.2139 | 0.2085 | 0.1806 | 0.2696 | 0.2625 | **0.0218** |
| | $\overline{AUPRC}$ - MD | 0.2606 | 0.2381 | 0.2297 | 0.2035 | 0.2937 | 0.2874 | **0.0168** |
| | CE - MD | 0.1540 | 0.1839 | 0.1689 | 0.1572 | 0.1487 | 0.1446 | **0.0234** |
| | MR - MD | 0.2967 | 0.2652 | 0.2620 | 0.2516 | 0.3101 | 0.2999 | **0.0468** |
| | $\overline{F_1}$ - MD | 0.2848 | 0.2195 | 0.2534 | 0.2613 | 0.2973 | 0.2975 | **0.0502** |
| | $\overline{AUROC}$ - EMD | 0.2736 | 0.2472 | 0.2449 | 0.2155 | 0.2897 | 0.2841 | **0.1121** |
| | $\overline{AUPRC}$ - EMD | 0.2653 | 0.2451 | 0.2429 | 0.2176 | 0.2852 | 0.2812 | **0.0912** |
| | CE - EMD | 0.2083 | 0.2318 | 0.2355 | 0.2147 | 0.2055 | 0.2003 | **0.1159** |
| | MR - MD | 0.3409 | 0.3162 | 0.3258 | 0.3026 | 0.3442 | 0.3388 | **0.1872** |
| | $\overline{F_1}$ - MD | 0.3237 | 0.2756 | 0.3031 | 0.3008 | 0.3215 | 0.3271 | **0.1779** |

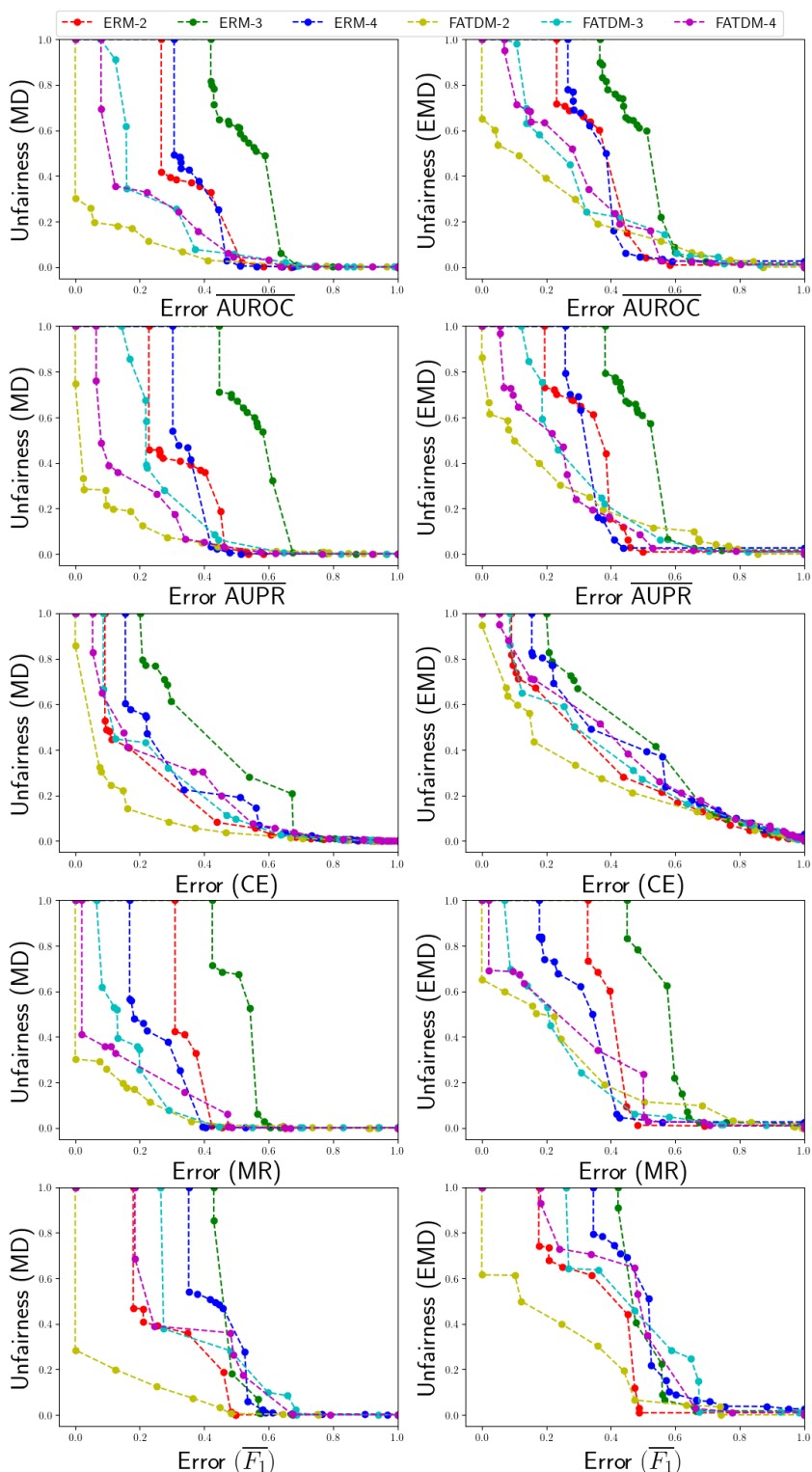

Figure 11: Error-unfairness curves with respect to equalized odds of FATDM and ERM on Edema disease dataset when training with different numbers of source domains. Names in the figure legend are in the form of X-Y where X is the model and Y is the number of source domains (e.g., ERM-2 means training ERM on two source domains.)

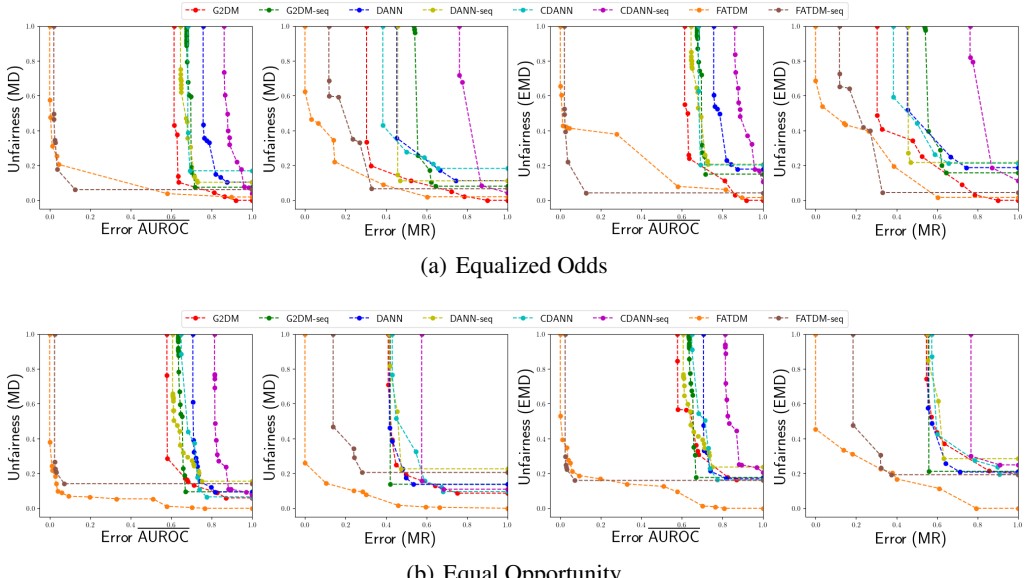

(a) Equalized Odds

(b) Equal Opportunity

Figure 12: Fairness-accuracy trade-off (Pareto frontier) of models trained with simultaneous and sequential (i.e., models with '-seq' suffix) approaches, and `FATDM-CycleGAN` (i.e., use `CycleGAN` instead of `StarGAN` as density mapping functions) on Cardiomegaly disease dataset: error-unfairness curves are constructed by varying $\omega \in [0, 10]$ and the values of error and unfairness are normalized to $[0, 1]$. Lower-left points and the smaller area under the curve indicate the model has a better fairness-accuracy trade-off (Pareto optimality).

# D    ADDITIONAL RESULTS & LEMMAS

## D.1    TIGHTER UPPER BOUND FOR ACCURACY

**Corollary 5.1** *We can replace **term (ii)** in Thm. 1 with the following term to attain a tighter upper bound for accuracy:*

$$\sqrt{2}C \min_{i \in [N]} \left( d_{JS}\left(P_{D^T}^Y, P_{D_i^S}^Y\right) + \sqrt{2\eta_{TV}\mathbb{E}_{z \sim P_{D_i^T}(z)}\left[d_{JS}\left(P_{D^T}^{X|Y}, P_{D_i^T}^{X|Y}\right)^2\right]} \right).$$

*where* $\eta_{TV} = \sup\limits_{P_{D_i}^X \neq P_{D_j}^X} \dfrac{\mathcal{D}_{TV}\left(P_{D_i}^Z, P_{D_j}^Z\right)}{\mathcal{D}_{TV}\left(P_{D_i}^X, P_{D_j}^X\right)} \leq 1$ *is called Dobrushin's coefficient (Polyanskiy & Wu,*

*2017).*

This result suggests that we can further optimize **term (ii)** in Thm. 1 by minimizing $\eta_{TV}$. It has been shown in Shui et al. (2022) that $\eta_{TV}$ can be controlled by Lipschitz constant of the feature mapping $g : \mathcal{X} \to \mathcal{Z}$ when $g$ follows Gaussian distribution. The Lipschitz constant of $g$, in turn, can be upper bounded by the Frobenius norm of Jacobian matrix with respect to $g$ (Miyato et al., 2018). However, in practice, we found that computing Jacobian matrix of $g$ is computationally expensive when dimension of representation $Z$ is large, and optimizing it together with invariant constraints does not improve the performances of models in our experiments.

## D.2    LEMMAS FOR PROVING THEOREM 1

**Lemma 6** *Let $X$ be the random variable in domains $D_i$ and $D_j$, and $\mathcal{E}$ be an event that $P_{D_j}^X \geq P_{D_i}^X$, then we have:*

$$\int_{\mathcal{E}} \left|P_{D_j}^X - P_{D_i}^X\right| dX = \int_{\overline{\mathcal{E}}} \left|P_{D_j}^X - P_{D_i}^X\right| dX = \frac{1}{2} \int \left|P_{D_j}^X - P_{D_i}^X\right| dX$$

*where $\overline{\mathcal{E}}$ is the complement of event $\mathcal{E}$.*

**Lemma 7** *Let $X$ be the random variable in domains $D_i$ and $D_j$, let $f : \mathcal{X} \to \mathbb{R}_+$ be a non-negative function bounded by $C$, then we have:*

$$\mathbb{E}_{D_j}[f(X)] - \mathbb{E}_{D_i}[f(X)] \leq \frac{C}{\sqrt{2}} \sqrt{\min \left( \mathcal{D}_{KL} \left( P_{D_i}^X \parallel P_{D_j}^X \right), \mathcal{D}_{KL} \left( P_{D_j}^X \parallel P_{D_i}^X \right) \right)}$$

*where $\mathcal{D}_{KL}(\cdot \parallel \cdot)$ is the KL-divergence between two distributions.*

**Lemma 8** *Suppose loss function $\mathcal{L}$ is upper bounded by $C$ and consider a classifier $\widehat{f} : \mathcal{X} \to \mathcal{Y}$. the expected classification error of $\widehat{f}$ in domain $D_j$ can be upper bounded by its error in domain $D_i$:*

$$\epsilon_{D_j}^{Acc} \left( \widehat{f} \right) \leq \epsilon_{D_i}^{Acc} \left( \widehat{f} \right) + \sqrt{2} C d_{JS} \left( P_{D_j}^{X,Y}, P_{D_i}^{X,Y} \right)$$

*where $X, Y$ are random variables denoting feature and label in domains $D_i$ and $D_j$.*

**Lemma 9** *Consider two distributions $P_{D_i}^X$ and $P_{D_j}^X$ over $\mathcal{X}$. Let $P_{D_i}^Z$ and $P_{D_j}^Z$ be the induced distributions over $\mathcal{Z}$ by mapping function $g : \mathcal{X} \to \mathcal{Z}$, then we have:*

$$d_{JS}(P_{D_i}^X, P_{D_j}^X) \geq d_{JS}(P_{D_i}^Z, P_{D_j}^Z)$$

**Lemma 10** *(Phung et al., 2021) Consider domain $D$ with joint distribution $P_D^{X,Y}$ and labeling function $f_D : \mathcal{X} \to \mathcal{Y}^\Delta$ from feature space to label space. Given mapping function $g : \mathcal{X} \to \mathcal{Z}$ from feature to representation space, we define labeling function $h_D : \mathcal{Z} \to \mathcal{Y}^\Delta$ from representation space to label space as $h_D(Z)_Y = f_D(X)_Y \circ g^{-1}(Z) = \frac{\int_{g^{-1}(Z)} f_D(X)_Y P_D^X dX}{\int_{g^{-1}(Z)} P_D^X dX}$. Similarly, let $\widehat{f}$ be the hypothesis from feature space, then the corresponding hypothesis $\widehat{h}$ from representation space under the mapping function $g$ is computed as $\widehat{h}(Z)_Y = \frac{\int_{g^{-1}(Z)} \widehat{f}(X)_Y P_D^X dX}{\int_{g^{-1}(Z)} P_D^X dX}$. Let $\epsilon_D^{Acc}(\widehat{f}) = \mathbb{E}_D \left[ \mathcal{L}(\widehat{f}(X), Y) \right]$ and $\epsilon_D^{Acc}(\widehat{h}) = \mathbb{E}_D \left[ \mathcal{L}(\widehat{h}(Z), Y) \right]$ be expected errors defined with respect to feature space and representation space, respectively. We have:*

$$\epsilon_D^{Acc} \left( \widehat{f} \right) = \epsilon_D^{Acc} \left( \widehat{h} \right)$$

### D.3 Lemmas for proving Corollary 1.1

**Lemma 11** *Consider two random variables $X, Y$. Let $P_{D_i}^{X,Y}, P_{D_j}^{X,Y}$ be two joint distributions defined in domains $D_i$ and $D_j$, respectively. Then, JS-divergence $\mathcal{D}_{JS} \left( P_{D_i}^{X,Y} \parallel P_{D_j}^{X,Y} \right)$ and KL-divergence $\mathcal{D}_{KL} \left( P_{D_i}^{X,Y} \parallel P_{D_j}^{X,Y} \right)$ can be decomposed as follows:*

$$\mathcal{D}_{KL} \left( P_{D_i}^{X,Y} \parallel P_{D_j}^{X,Y} \right) = \mathcal{D}_{KL} \left( P_{D_i}^{Y} \parallel P_{D_j}^{Y} \right) + \mathbb{E}_{D_i} \left[ \mathcal{D}_{KL} \left( P_{D_i}^{X|Y} \parallel P_{D_j}^{X|Y} \right) \right]$$
$$\mathcal{D}_{JS} \left( P_{D_i}^{X,Y} \parallel P_{D_j}^{X,Y} \right) \leq \mathcal{D}_{JS} \left( P_{D_i}^{Y} \parallel P_{D_j}^{Y} \right) + \mathbb{E}_{D_i} \left[ \mathcal{D}_{JS} \left( P_{D_i}^{X|Y} \parallel P_{D_j}^{X|Y} \right) \right]$$
$$+ \mathbb{E}_{D_j} \left[ \mathcal{D}_{JS} \left( P_{D_i}^{X|Y} \parallel P_{D_j}^{X|Y} \right) \right]$$

### D.4 Lemmas for proving Theorem 2

**Lemma 12** *Under Assumption in Theorem 2, the following holds for any domain $D$:*

$$\sqrt{\epsilon_D^{Acc}(\widehat{f})} = \sqrt{\mathbb{E}_D[\mathcal{L}(\widehat{f}(X), Y)]} \geq \sqrt{\frac{2c}{|\mathcal{Y}|}} d_{JS}(P_D^Y, P_D^{\widehat{Y}})^2, \ \forall \widehat{f}$$

*where $\widehat{Y}$ is the prediction made by randomized predictor $\widehat{f}$.*

## D.5 LEMMAS FOR PROVING THEOREM 3

**Definition 13** *Given domain $D_i$ with binary random variable $A$ denoting the sensitive attribute, the unfairness measures that evaluate the violation of equalized odd (EO) and equal opportunity (EP) criteria between sensitive groups of this domain are defined as follows.*

$$\epsilon_{D_i}^{EO}\left(\widehat{f}\right) = \left|R_{D_i}^{0,0}\left(\widehat{f}\right) - R_{D_i}^{0,1}\left(\widehat{f}\right)\right| + \left|R_{D_i}^{1,0}\left(\widehat{f}\right) - R_{D_i}^{1,1}\left(\widehat{f}\right)\right|$$

$$\epsilon_{D_i}^{EP}\left(\widehat{f}\right) = \left|R_{D_i}^{1,0}\left(\widehat{f}\right) - R_{D_i}^{1,1}\left(\widehat{f}\right)\right|$$

*where $R_{D_i}^{y,a}\left(\widehat{f}\right) = \mathbb{E}_{D_i}\left[\widehat{f}(X)_1 | Y = y, A = a\right]$.*

**Lemma 14** *Given two domains $D_i$ and $D_j$, under Definition 13, $R_{D_j}^{y,a}\left(\widehat{f}\right)$ can be bounded by $R_{D_i}^{y,a}\left(\widehat{f}\right)$ as follows.*

$$R_{D_j}^{y,a}\left(\widehat{f}\right) \leq R_{D_i}^{y,a}\left(\widehat{f}\right) + \sqrt{2}d_{JS}\left(P_{D_j}^{X|Y=y,A=a}, P_{D_i}^{X|Y=y,A=a}\right) \quad \forall y, a \in \{0,1\}$$

**Lemma 15** *Given two domains $D_i$ and $D_j$, under Definition 13, the unfairness in domain $D_j$ can be upper bounded by the unfairness measure in domain $D_i$ as follows.*

$$\epsilon_{D_j}^{EO}\left(\widehat{f}\right) \leq \epsilon_{D_i}^{EO}\left(\widehat{f}\right) + \sqrt{2}\sum_{y=0,1}\sum_{a=0,1} d_{JS}\left(P_{D_j}^{X|Y=y,A=a}, P_{D_i}^{X|Y=y,A=a}\right)$$

$$\epsilon_{D_j}^{EP}\left(\widehat{f}\right) \leq \epsilon_{D_i}^{EP}\left(\widehat{f}\right) + \sqrt{2}\sum_{a=0,1} d_{JS}\left(P_{D_j}^{X|Y=1,A=a}, P_{D_i}^{X|Y=1,A=a}\right)$$

**Lemma 16** *Consider domain $D$ with distribution $P_D^{X,Y}$ and labeling function $f_D : \mathcal{X} \to \mathcal{Y}^\Delta$. Given mapping function $g : \mathcal{X} \to \mathcal{Z}$ from feature to representation space, we define labeling function $h_D : \mathcal{Z} \to \mathcal{Y}^\Delta$ from representation space to label space as $h_D(Z)_Y = f_D(X)_Y \circ g^{-1}(Z) = \frac{\int_{g^{-1}(Z)} f_D(X)_Y P_D^X dX}{\int_{g^{-1}(Z)} P_D^X dX}$. Similarly, let $\widehat{f}$ be the hypothesis from feature space, then the corresponding hypothesis $\widehat{h}$ from representation space under the mapping function $g$ is computed as $\widehat{h}(Z)_Y = \frac{\int_{g^{-1}(Z)} \widehat{f}(X)_Y P_D^X dX}{\int_{g^{-1}(Z)} P_D^X dX}$. Under Definition 13, we have:*

$$\epsilon_D^{EO}\left(\widehat{f}\right) = \epsilon_D^{EO}\left(\widehat{h}\right)$$

$$\epsilon_D^{EP}\left(\widehat{f}\right) = \epsilon_D^{EP}\left(\widehat{h}\right)$$

## D.6 LEMMAS FOR PROVING THEOREM 5

**Lemma 17** *Consider two domains $D_i$ and $D_j$, if there exist invertible mappings $m_{i,j}^y$ and $m_{i,j}^{y,a}$ such that $P_{D_i}^{X|y} = P_{D_j}^{m_{i,j}^y(X)|y}$ and $P_{D_i}^{X|y,a} = P_{D_j}^{m_{i,j}^{y,a}(X)|y,a}$, $\forall y \in \mathcal{Y}, a \in \mathcal{A}$, then $\mathcal{D}_{JS}\left(P_{D_i}^{Z|y} \parallel P_{D_j}^{Z|y}\right)$ and $\mathcal{D}_{JS}\left(P_{D_i}^{Z|y,a} \parallel P_{D_j}^{Z|y,a}\right)$ can be upper bounded by $\int_x P_{D_i}^{x|y}\mathcal{D}_{JS}\left(P^{Z|x} \parallel P^{Z|m_{i,j}^y(x)}\right) dx$ and $\int_x P_{D_i}^{x|y,a}\mathcal{D}_{JS}\left(P^{Z|x} \parallel P^{Z|m_{i,j}^{y,a}(x)}\right) dx$, respectively.*

## E PROOFS

### E.1 PROOFS OF THEOREMS

**Proof of Theorem 1.** First, we get the upper bound based on the representation space $\mathcal{Z}$. Then, we relate it with the feature space $\mathcal{X}$. Let $D_*^S \in \{D_i^S\}_{i=1}^N$ be the source domain that's nearest to the

target domain $D^T$. According to Lemma 8, we have upper bound of the expected classification error for the target domain based on each of the source domain as follows.

$$\epsilon_{D^T}^{\text{Acc}}\left(\widehat{h}\right) \leq \epsilon_{D_i^S}^{\text{Acc}}\left(\widehat{h}\right) + \sqrt{2}C d_{JS}\left(P_{D^T}^{Z,Y}, P_{D_i^S}^{Z,Y}\right) \quad \forall i \in [N]$$

Taking average of upper bounds based on all source domains, we have:

$$\begin{aligned}
\epsilon_{D^T}^{\text{Acc}}\left(\widehat{h}\right) &\leq \frac{1}{N}\sum_{i=1}^N \epsilon_{D_i^S}^{\text{Acc}}\left(\widehat{h}\right) + \frac{\sqrt{2}C}{N}\sum_{i=1}^N d_{JS}\left(P_{D^T}^{Z,Y}, P_{D_i^S}^{Z,Y}\right) \\
&\stackrel{(1)}{\leq} \frac{1}{N}\sum_{i=1}^N \epsilon_{D_i^S}^{\text{Acc}}\left(\widehat{h}\right) + \frac{\sqrt{2}C}{N}\sum_{i=1}^N d_{JS}\left(P_{D^T}^{Z,Y}, P_{D_*^S}^{Z,Y}\right) + \frac{\sqrt{2}C}{N}\sum_{i=1}^N d_{JS}\left(P_{D_*^S}^{Z,Y}, P_{D_i^S}^{Z,Y}\right) \\
&\stackrel{(2)}{\leq} \frac{1}{N}\sum_{i=1}^N \epsilon_{D_i^S}^{\text{Acc}}\left(\widehat{h}\right) + \sqrt{2}C \min_{i \in [N]} d_{JS}\left(P_{D^T}^{Z,Y}, P_{D_i^S}^{Z,Y}\right) + \sqrt{2}C \max_{i,j \in [N]} d_{JS}\left(P_{D_i^S}^{Z,Y}, P_{D_j^S}^{Z,Y}\right)
\end{aligned}$$

$$(7)$$

Here we have $\stackrel{(1)}{\leq}$ by using triangle inequality for JS-distance: $d_{JS}(P,R) \leq d_{JS}(P,Q) + d_{JS}(Q,R)$ with $P, Q,$ and $R = P_{D^T}, P_{D_*^S}$ and $P_{D_i^S}$, respectively. We have $\stackrel{(2)}{\leq}$ because $D_*^S \in \{D_i^S\}_{i=1}^N$ then $d_{JS}\left(P_{D_*^S}^{Z,Y}, P_{D_i^S}^{Z,Y}\right) \leq \max_{i,j \in [N]} d_{JS}\left(P_{D_i^S}^{Z,Y}, P_{D_j^S}^{Z,Y}\right)$. Similarly, we can obtain the upper bound based on the feature space $\mathcal{X}$ as follows.

$$\epsilon_{D^T}^{\text{Acc}}\left(\widehat{f}\right) \leq \frac{1}{N}\sum_{i=1}^N \epsilon_{D_i^S}^{\text{Acc}}\left(\widehat{f}\right) + \sqrt{2}C \min_{i \in [N]} d_{JS}\left(P_{D^T}^{X,Y}, P_{D_i^S}^{X,Y}\right) + \sqrt{2}C \max_{i,j \in [N]} d_{JS}\left(P_{D_i^S}^{X,Y}, P_{D_j^S}^{X,Y}\right)$$

$$(8)$$

However, the bounds in Eq. (7) and Eq. (8) are based on either feature space or representation space, which is not readily to use for practical algorithmic design because the actual objective is to minimize $\epsilon_{D^T}^{\text{Acc}}\left(\widehat{f}\right)$ in feature space by controlling $Z$ in representation space. According to Lemmas 9 and 10, we can derive the bound that relates feature and representation spaces as follows.

$$\begin{aligned}
\epsilon_{D^T}^{\text{Acc}}\left(\widehat{f}\right) &= \epsilon_{D^T}^{\text{Acc}}\left(\widehat{h}\right) \\
&\leq \frac{1}{N}\sum_{i=1}^N \epsilon_{D_i^S}^{\text{Acc}}\left(\widehat{h}\right) + \sqrt{2}C \min_{i \in [N]} d_{JS}\left(P_{D^T}^{Z,Y}, P_{D_i^S}^{Z,Y}\right) + \sqrt{2}C \max_{i,j \in [N]} d_{JS}\left(P_{D_i^S}^{Z,Y}, P_{D_j^S}^{Z,Y}\right) \\
&\leq \frac{1}{N}\sum_{i=1}^N \epsilon_{D_i^S}^{\text{Acc}}\left(\widehat{f}\right) + \sqrt{2}C \min_{i \in [N]} d_{JS}\left(P_{D^T}^{X,Y}, P_{D_i^S}^{X,Y}\right) + \sqrt{2}C \max_{i,j \in [N]} d_{JS}\left(P_{D_i^S}^{Z,Y}, P_{D_j^S}^{Z,Y}\right)
\end{aligned}$$

$$(9)$$

**Proof of Corollary 1.1.**

$$\begin{aligned}
d_{JS}\left(P_{D_i^S}^{Z,Y}, P_{D_j^S}^{Z,Y}\right) &= \sqrt{\mathcal{D}_{JS}\left(P_{D_i^S}^{Z,Y} \| P_{D_j^S}^{Z,Y}\right)} \\
&\stackrel{(1)}{\leq} \sqrt{\mathcal{D}_{JS}\left(P_{D_i^S}^Y \| P_{D_j^S}^Y\right) + 2\mathbb{E}_{z \sim P_{D_{i,j}^S}(z)}\left[\mathcal{D}_{JS}\left(P_{D_i^S}^{Z|Y} \| P_{D_j^S}^{Z|Y}\right)\right]} \\
&\stackrel{(2)}{\leq} d_{JS}\left(P_{D_i^S}^Y, P_{D_j^S}^Y\right) + \sqrt{2\mathbb{E}_{z \sim P_{D_{i,j}^S}(z)}\left[d_{JS}\left(P_{D_i^S}^{Z|Y}, P_{D_j^S}^{Z|Y}\right)^2\right]}
\end{aligned}$$

Here we have $\stackrel{(1)}{\leq}$ by using Lemma 11 to decompose the JS-divergence of the joint distributions and $\stackrel{(2)}{\leq}$ by using inequality $\sqrt{a+b} \leq \sqrt{a} + \sqrt{b}$.

This new upper bound, combined with Thm. 1 suggests learning representation $Z$ such that $P_{D_i^S}^{Z|Y}$ is invariant across source domains, or in another word, $Z \perp D \mid Y$. This result is consistent with Thm. 4: when the target domain $D^T$ is the mixture of source domains $\{D_i^S\}_{i=1}^N$, and when $P_{D_i^S}^Y$ and $P_{D_i^S}^{Z|Y}$ are invariant across source domains, we have $d_{JS}\left(P_{D^T}^{Z,Y}, P_{D_i^S}^{Z,Y}\right) = d_{JS}\left(P_{D_i^S}^{Z,Y}, P_{D_j^S}^{Z,Y}\right) = 0$, implying $\epsilon_{D^T}^{\text{Acc}}\left(\widehat{f}\right) \leq \frac{1}{N}\sum_{i=1}^N \epsilon_{D_i^S}^{\text{Acc}}\left(\widehat{f}\right) = \epsilon_{D_i^S}^{\text{Acc}}\left(\widehat{f}\right) \forall i \in [N]$.

**Proof of Corollary 5.1 (tighter upper bound for accuracy).** The bound in Eq. (9) is constructed using Lemma 9. Indeed, we can make this bound tighter using the strong data processing inequality for JS-divergence (Polyanskiy & Wu, 2017), as stated below.

$$\mathcal{D}_{JS}\left(P_{D_i}^Z \parallel P_{D_j}^Z\right) \leq \eta_{JS}\mathcal{D}_{JS}\left(P_{D_i}^X \parallel P_{D_j}^X\right) \leq \eta_{TV}\mathcal{D}_{JS}\left(P_{D_i}^X \parallel P_{D_j}^X\right)$$

where $Z$ is random variable induced from random variable $X$, and $P_{D_i}^X$ and $P_{D_i}^X$ are two distribution over $\mathcal{X}$, and $\eta_{JS} = \sup_{P_{D_i}^X \neq P_{D_j}^X} \frac{\mathcal{D}_{JS}\left(P_{D_i}^Z, P_{D_j}^Z\right)}{\mathcal{D}_{JS}\left(P_{D_i}^X, P_{D_j}^X\right)} \leq \eta_{TV} = \sup_{P_{D_i}^X \neq P_{D_j}^X} \frac{\mathcal{D}_{TV}\left(P_{D_i}^Z, P_{D_j}^Z\right)}{\mathcal{D}_{TV}\left(P_{D_i}^X, P_{D_j}^X\right)} \leq 1$, $\mathcal{D}_{TV}$ is the total variation distance. $\eta_{TV}$ is called the Dobrushin's coefficient (Polyanskiy & Wu, 2017).

Apply Lemma 11 and this inequality to the second term in the right hand side of Eq. (7) (similar to the proof of Corollary 1.1), we have:

$$\sqrt{2}C \min_{i \in [N]} d_{JS}\left(P_{D^T}^{Z,Y}, P_{D_i^S}^{Z,Y}\right)$$

$$\leq \sqrt{2}C \min_{i \in [N]} \left(d_{JS}\left(P_{D^T}^Y, P_{D_i^S}^Y\right) + \sqrt{2\mathbb{E}_{z \sim P_{D_i^T}(z)}\left[d_{JS}\left(P_{D^T}^{Z|Y}, P_{D_i^T}^{Z|Y}\right)^2\right]}\right)$$

$$\leq \sqrt{2}C \min_{i \in [N]} \left(d_{JS}\left(P_{D^T}^Y, P_{D_i^S}^Y\right) + \sqrt{2\eta_{TV}\mathbb{E}_{z \sim P_{D_i^T}(z)}\left[d_{JS}\left(P_{D^T}^{X|Y}, P_{D_i^T}^{X|Y}\right)^2\right]}\right) \quad (10)$$

**Proof of Theorem 2.** Consider a source domain $D_i^S$ and target domain $D^T$. Because JS-distance $d_{JS}(\cdot, \cdot)$ is a distance metric, we have triangle inequality:

$$d_{JS}(P_{D_i^S}^Y, P_{D^T}^Y) \leq d_{JS}(P_{D_i^S}^Y, P_{D_i^S}^{\widehat{Y}}) + d_{JS}(P_{D_i^S}^{\widehat{Y}}, P_{D^T}^{\widehat{Y}}) + d_{JS}(P_{D^T}^{\widehat{Y}}, P_{D^T}^Y)$$

Since $X \xrightarrow{g} Z \xrightarrow{\widehat{h}} \widehat{Y}$, we have $d_{JS}(P_{D_i^S}^{\widehat{Y}}, P_{D^T}^{\widehat{Y}}) \leq d_{JS}(P_{D_i^S}^Z, P_{D^T}^Z)$. Using Lemma 12, the following holds when $d_{JS}(P_{D_i^S}^Y, P_{D^T}^Y) \geq d_{JS}(P_{D_i^S}^Z, P_{D^T}^Z)$

$$\begin{aligned}
\left(d_{JS}(P_{D_i^S}^Y, P_{D^T}^Y) - d_{JS}(P_{D_i^S}^Z, P_{D^T}^Z)\right)^2 &\leq \left(d_{JS}(P_{D_i^S}^Y, P_{D_i^S}^{\widehat{Y}}) + d_{JS}(P_{D^T}^{\widehat{Y}}, P_{D^T}^Y)\right)^2 \\
&\leq 2\left(d_{JS}(P_{D_i^S}^Y, P_{D_i^S}^{\widehat{Y}})^2 + d_{JS}(P_{D^T}^{\widehat{Y}}, P_{D^T}^Y)^2\right) \\
&\leq \frac{2}{\sqrt{\frac{2c}{|\mathcal{Y}|}}}\left(\sqrt{\epsilon_{D_i^S}^{\text{Acc}}(\widehat{f})} + \sqrt{\epsilon_{D^T}^{\text{Acc}}(\widehat{f})}\right) \\
&\leq \sqrt{\frac{4|\mathcal{Y}|}{c}\left(\epsilon_{D_i^S}^{\text{Acc}}(\widehat{f}) + \epsilon_{D^T}^{\text{Acc}}(\widehat{f})\right)}
\end{aligned}$$

The last inequality is by AM-GM inequality.

Therefore, when $d_{JS}(P_{D_i^S}^Y, P_{D^T}^Y) \geq d_{JS}(P_{D_i^S}^Z, P_{D^T}^Z)$, we have

$$\epsilon_{D_i^S}^{\text{Acc}}(\widehat{f}) + \epsilon_{D^T}^{\text{Acc}}(\widehat{f}) \geq \frac{c}{4|\mathcal{Y}|}\left(d_{JS}(P_{D_i^S}^Y, P_{D^T}^Y) - d_{JS}(P_{D_i^S}^Z, P_{D^T}^Z)\right)^4$$

The above holds for any source domain $D_i^S$. Average over all $N$ source domains, we have

$$\frac{1}{N}\sum_{i=1}^N \epsilon_{D_i^S}^{\text{Acc}}(\widehat{f}) + \epsilon_{D^T}^{\text{Acc}}(\widehat{f}) \geq \frac{c}{4|\mathcal{Y}|N}\sum_{i=1}^N \left(d_{JS}(P_{D_i^S}^Y, P_{D^T}^Y) - d_{JS}(P_{D_i^S}^Z, P_{D^T}^Z)\right)^4$$

**Proof of Theorem 3.** The proof is based on Lemmas 15 and 16 and similar to the proof of Thm. 1.

Let $D_*^S \in \{D_i^S\}_{i=1}^N$ be the source domain nearest to the target domain $D^T$. According to Lemma 15, we have upper bound of the unfairness measured with respect to the representation space for the target domain based on each of the source domain. For equal opportunity (EP), we have:

$$\epsilon_{D^T}^{\text{EP}}\left(\widehat{h}\right) \leq \epsilon_{D_i^S}^{\text{EP}}\left(\widehat{h}\right) + \sqrt{2}\sum_{a\in\{0,1\}} d_{JS}\left(P_{D^T}^{Z|Y=1,A=a}, P_{D_i^S}^{Z|Y=1,A=a}\right)$$

Taking average of upper bounds based on all source domains, we have:

$$\begin{aligned}
\epsilon_{D^T}^{\text{EP}}\left(\widehat{h}\right) &\leq \frac{1}{N}\sum_{i=1}^N \epsilon_{D_i^S}^{\text{EP}}\left(\widehat{h}\right) + \frac{\sqrt{2}}{N}\sum_{i=1}^N\sum_{a\in\{0,1\}} d_{JS}\left(P_{D^T}^{Z|Y=1,A=a}, P_{D_i^S}^{Z|Y=1,A=a}\right) \\
&\leq \frac{1}{N}\sum_{i=1}^N \epsilon_{D_i^S}^{\text{EP}}\left(\widehat{h}\right) + \frac{\sqrt{2}}{N}\sum_{i=1}^N\sum_{a\in\{0,1\}} d_{JS}\left(P_{D^T}^{Z|Y=1,A=a}, P_{D_*^S}^{Z|Y=1,A=a}\right) \\
&\quad + \frac{\sqrt{2}}{N}\sum_{i=1}^N\sum_{a\in\{0,1\}} d_{JS}\left(P_{D_*^S}^{Z|Y=1,A=a}, P_{D_i^S}^{Z|Y=1,A=a}\right) \\
&\leq \frac{1}{N}\sum_{i=1}^N \epsilon_{D_i^S}^{\text{EP}}\left(\widehat{h}\right) + \sqrt{2}\min_{i\in[N]}\sum_{a\in\{0,1\}} d_{JS}\left(P_{D^T}^{Z|Y=1,A=a}, P_{D_i^S}^{Z|Y=1,A=a}\right) \\
&\quad + \sqrt{2}\max_{i,j\in[N]}\sum_{a\in\{0,1\}} d_{JS}\left(P_{D_i^S}^{Z|Y=1,A=a}, P_{D_j^S}^{Z|Y=1,A=a}\right)
\end{aligned}$$

According to Lemmas 9 and 16. we can relate this bound to the feature space as follows.

$$\begin{aligned}
\epsilon_{D^T}^{\text{EP}}\left(\widehat{f}\right) &= \epsilon_{D^T}^{\text{EP}}\left(\widehat{h}\right) \\
&\leq \frac{1}{N}\sum_{i=1}^N \epsilon_{D_i^S}^{\text{EP}}\left(\widehat{h}\right) + \sqrt{2}\min_{i\in[N]}\sum_{a\in\{0,1\}} d_{JS}\left(P_{D^T}^{Z|Y=1,A=a}, P_{D_i^S}^{Z|Y=1,A=a}\right) \\
&\quad + \sqrt{2}\max_{i,j\in[N]}\sum_{a\in\{0,1\}} d_{JS}\left(P_{D_i^S}^{Z|Y=1,A=a}, P_{D_j^S}^{Z|Y=1,A=a}\right) \\
&\leq \frac{1}{N}\sum_{i=1}^N \epsilon_{D_i^S}^{\text{EP}}\left(\widehat{f}\right) + \sqrt{2}\min_{i\in[N]}\sum_{a\in\{0,1\}} d_{JS}\left(P_{D^T}^{X|Y=1,A=a}, P_{D_i^S}^{X|Y=1,A=a}\right) \\
&\quad + \sqrt{2}\max_{i,j\in[N]}\sum_{a\in\{0,1\}} d_{JS}\left(P_{D_i^S}^{Z|Y=1,A=a}, P_{D_j^S}^{Z|Y=1,A=a}\right)
\end{aligned}$$

Similarly, we got the upper bound for unfairness measure with respect to equalized odds as follows.

$$\begin{aligned}
\epsilon_{D^T}^{\text{EO}}\left(\widehat{f}\right) &\leq \frac{1}{N}\sum_{i=1}^N \epsilon_{D_i^S}^{\text{EO}}\left(\widehat{f}\right) + \sqrt{2}\min_{i\in[N]}\sum_{y\in\{0,1\}}\sum_{a\in\{0,1\}} d_{JS}\left(P_{D^T}^{X|Y=y,A=a}, P_{D_i^S}^{X|Y=y,A=a}\right) \\
&\quad + \sqrt{2}\max_{i,j\in[N]}\sum_{y\in\{0,1\}}\sum_{a\in\{0,1\}} d_{JS}\left(P_{D_i^S}^{Z|Y=y,A=a}, P_{D_j^S}^{Z|Y=y,A=a}\right)
\end{aligned} \tag{11}$$

**Proof of Theorem 4.** Consider two source domains, $D_i^S$ and $D_j^S$, if $P_{D_i^S}^Y = P_{D_j^S}^Y$, we can learn the mapping function $g = P_\theta\left(Z|X\right)$ such that $P_{D_i^S}^{Z|Y} = P_{D_j^S}^{Z|Y}$. Note that this mapping function always exists. In particular, the trivial solution for $Z$ that satisfies $P_{D_i^S}^{Z|Y} = P_{D_j^S}^{Z|Y}$ is making $Z \perp Y, D$ (e.g.,

$P_\theta \left( Z | X \right) = \mathcal{N} \left( \mathbf{0}, \mathbf{I} \right)$). Then we have:

$$\epsilon_{D_i^S}^{\text{Acc}} \left( \widehat{h} \right) = \mathbb{E}_{z \sim P_{D_i^S}^Z, y \sim h_{D_i^S}(z)} \left[ \mathcal{L} \left( \widehat{h}(Z), Y \right) \right]$$

$$= \mathbb{E}_{y \sim P_{D_i^S}^Y, z \sim P_{D_i^S}^{Z|Y}} \left[ \mathcal{L} \left( \widehat{h}(Z), Y \right) \right]$$

$$= \mathbb{E}_{y \sim P_{D_j^S}^Y, z \sim P_{D_j^S}^{Z|Y}} \left[ \mathcal{L} \left( \widehat{h}(Z), Y \right) \right]$$

$$= \mathbb{E}_{z \sim P_{D_j^S}^Z, y \sim h_{D_j^S}(z)} \left[ \mathcal{L} \left( \widehat{h}(Z), Y \right) \right]$$

$$= \epsilon_{D_j^S}^{\text{Acc}} \left( \widehat{h} \right)$$

For unseen target domain $D^T$ in $\Lambda$, we have:

$$\epsilon_{D^T}^{\text{Acc}} \left( \widehat{h} \right) = \mathbb{E}_{D^T} \left[ \mathcal{L} \left( \widehat{h}(Z), Y \right) \right]$$

$$= \int_{\mathcal{Z} \times \mathcal{Y}} \mathcal{L} \left( \widehat{h}(Z), Y \right) P_{D^T}^{Y,Z} \, dY \, dZ$$

$$= \int_{\mathcal{Z} \times \mathcal{Y}} \mathcal{L} \left( \widehat{h}(Z), Y \right) \sum_{i=1}^N \pi_i P_{D_i^S}^{Y,Z} \, dY \, dZ$$

$$= \sum_{i=1}^N \pi_i \int_{\mathcal{Z} \times \mathcal{Y}} \mathcal{L} \left( \widehat{h}(Z), Y \right) P_{D_i^S}^{Y,Z} \, dY \, dZ$$

$$= \sum_{i=1}^N \pi_i \mathbb{E}_{D_i^S} \left[ \mathcal{L} \left( \widehat{h}(Z), Y \right) \right]$$

$$= \mathbb{E}_{D_i^S} \left[ \mathcal{L} \left( \widehat{h}(Z), Y \right) \right] \quad \forall i \in [N]$$

$$= \epsilon_{D_i^S}^{\text{Acc}} \left( \widehat{h} \right) \quad \forall i \in [N]$$

By Lemma 10, we have $\epsilon_{D^T}^{\text{Acc}} \left( \widehat{h} \right) = \epsilon_{D_i^S}^{\text{Acc}} \left( \widehat{h} \right) = \epsilon_{D^T}^{\text{Acc}} \left( \widehat{f} \right) = \epsilon_{D_i^S}^{\text{Acc}} \left( \widehat{f} \right)$.

For fairness, we only give the proof for equalized odds (EO), we can easily get the similar derivation for equal opportunity. For any $Z$ that satisfies $P_{D_i^S}^{Z|Y=y,A=a} = P_{D_j^S}^{Z|Y=y,A=a} \; \forall y, a \in \{0,1\}$, we have:

$$\epsilon_{D_i^S}^{\text{EO}} \left( \widehat{h} \right) = \sum_{y \in \{0,1\}} \mathcal{D} \left( P_{D_i^S}^{\widehat{h}(Z)_1 | Y=y, A=0} \| P_{D_i^S}^{\widehat{h}(Z)_1 | Y=y, A=1} \right)$$

$$= \sum_{y \in \{0,1\}} \mathcal{D} \left( P_{D_j^S}^{\widehat{h}(Z)_1 | Y=y, A=0} \| P_{D_j^S}^{\widehat{h}(Z)_1 | Y=y, A=1} \right)$$

$$= \epsilon_{D_j^S}^{\text{EO}} \left( \widehat{h} \right)$$

For unseen target domain $D^T$ in $\Lambda$, we have:

$$\epsilon_{D^T}^{\text{EO}} \left( \widehat{h} \right) = \sum_{y \in \{0,1\}} \mathcal{D} \left( P_{D^T}^{\widehat{h}(Z)_1 | Y=y, A=0} \| P_{D^T}^{\widehat{h}(Z)_1 | Y=y, A=1} \right)$$

$$= \sum_{y \in \{0,1\}} \mathcal{D} \left( \sum_{i=1}^N \pi_i P_{D_i^S}^{\widehat{h}(Z)_1 | Y=y, A=0} \| \sum_{i=1}^N \pi_i P_{D_i^S}^{\widehat{h}(Z)_1 | Y=y, A=1} \right)$$

$$= \sum_{y \in \{0,1\}} \mathcal{D} \left( P_{D_i^S}^{\widehat{h}(Z)_1 | Y=y, A=0} \| P_{D_i^S}^{\widehat{h}(Z)_1 | Y=y, A=1} \right) \quad \forall i \in [N]$$

$$= \epsilon_{D_i^S}^{\text{EO}} \left( \widehat{h} \right) \quad \forall i \in [N]$$

Similar to the proof of accuracy, $Z$ that satisfies $P_{D_i^S}^{Z|Y=y,A=a} = P_{D_j^S}^{Z|Y=y,A=a}$ $\forall y, a \in \{0,1\}, i, j \in [N]$ always exists. The trivial solution for is $Z$ that satisfies $Z \perp Y, A, D$.

By Lemma 16, we have $\epsilon_{D^T}^{\text{EO}}\left(\widehat{h}\right) = \epsilon_{D_i^S}^{\text{EO}}\left(\widehat{h}\right) = \epsilon_{D^T}^{\text{EO}}\left(\widehat{f}\right) = \epsilon_{D_i^S}^{\text{EO}}\left(\widehat{f}\right)$.

For equal opportunity (EP), $Z$ only need to satisfy the condition for positive label, i.e., $P_{D_i^S}^{Z|Y=1,A=a} = P_{D_j^S}^{Z|Y=1,A=a}$ $\forall a \in \{0,1\}, i, j \in [N]$.

**Proof of Theorem 5.** According to Lemma 17, we have:

$$\mathcal{D}_{JS}\left(P_{D_i}^{Z|y} \parallel P_{D_j}^{Z|y}\right) \leq \int_x P_{D_j}^{x|y} \mathcal{D}_{JS}\left(P_{D_i}^{Z|x} \parallel P_{D_i}^{Z|m_{i,j}^y(x)}\right) dx \tag{12}$$

Then, minimizing $D_{JS}\left(P_{D_i}^{Z|y} \parallel P_{D_j}^{Z|y}\right)$ can be achieved by minimizing $\mathcal{D}_{JS}\left(P^{Z|x} \parallel P^{Z|m_{i,j}^y(x)}\right)$ $\forall x \in \mathcal{X}$. We can upper bound $\mathcal{D}_{JS}\left(P^{Z|x} \parallel P^{Z|m_{i,j}^y(x)}\right)$ as follows

$$\mathcal{D}_{JS}\left(P^{Z|x} \parallel P^{Z|m_{i,j}^y(x)}\right) \leq \mathcal{D}_{TV}\left(P^{Z|x} \parallel P^{Z|m_{i,j}^y(x)}\right)$$
$$\leq \sqrt{2}\, d_{1/2}\left(P^{Z|x}, P^{Z|m_{i,j}^y(x)}\right)$$
$$\overset{(1)}{=} \sqrt{2}\, d_{1/2}\left(\mathcal{N}\left(\mu(x); \sigma^2 \mathbf{I}_d\right), \mathcal{N}\left(\mu\left(m_{i,j}^y(x)\right); \sigma^2 \mathbf{I}_d\right)\right) \tag{13}$$

where $\mathcal{D}_{TV}$ and $d_{1/2}$ are total variation distance and Hellinger distance between two distributions, respectively. We have $\overset{(1)}{=}$ because of our choice for representation mapping $g(x) := P^{Z|x} = \mathcal{N}\left(\mu(x); \sigma^2 \mathbf{I}_d\right)$. According to Devroye et al. (2018), the Hellinger distance between two multivariate normal distributions over $\mathbb{R}^d$ has a closed form as follows

$$d_{1/2}\left(\mathcal{N}\left(\mu_1; \Sigma_1\right), \mathcal{N}\left(\mu_2; \Sigma_2\right)\right)$$
$$= \sqrt{1 - \frac{\det\left(\Sigma_1\right)^{1/4} \det\left(\Sigma_2\right)^{1/4}}{\det\left(\frac{\Sigma_1+\Sigma_2}{2}\right)^{1/2}} \exp\left(-\frac{1}{8}\left(\mu_1 - \mu_2\right)^T \left(\frac{\Sigma_1 + \Sigma_2}{2}\right)^{-1}\left(\mu_1 + \mu_2\right)\right)} \tag{14}$$

where $\mu_1, \mu_2, \Sigma_1, \Sigma_2$ are mean vectors and covariance matrices of the two normal distributions. In Eq. (14), let $\mu_1 = \mu(x)$, $\mu_2 = \mu\left(m_{i,j}^y(x)\right)$, $\Sigma_1 = \Sigma_2 = \sigma^2 \mathbf{I}_d$, then we have:

$$d_{1/2}\left(\mathcal{N}\left(\mu(x); \sigma^2 \mathbf{I}_d\right), \mathcal{N}\left(\mu\left(m_{i,j}^y(x)\right); \sigma^2 \mathbf{I}_d\right)\right)$$
$$= \sqrt{1 - \exp\left(-\frac{1}{8d\sigma^2}\left(\mu(x) - \mu\left(m_{i,j}^y(x)\right)\right)^T \left(\mu(x) - \mu\left(m_{i,j}^y(x)\right)\right)\right)}$$
$$= \sqrt{1 - \exp\left(-\frac{1}{8d\sigma^2}\left\|\mu(x) - \mu\left(m_{i,j}^y(x)\right)\right\|_2^2\right)} \tag{15}$$

From Eq. (15), we can see that Helinger distance between two representation distributions $P^{Z|x}$ and $P^{Z|m_{i,j}^y(x)}$ is the function of their means $\mu(x)$ and $\mu\left(m_{i,j}^y(x)\right)$. Combining this with Eq. (12) and Eq. (13), we conclude that minimizing $d_{JS}\left(P_{D_i^S}^{Z|y}, P_{D_j^S}^{Z|y}\right)$ can be reduced to minimizing $\left\|\mu(x) - \mu\left(m_{i,j}^y(x)\right)\right\|_2$ which can be implemented as the mean square error between $\mu(x)$ and $\mu\left(m_{i,j}^y(x)\right)$ in practice. Proof for $d_{JS}\left(P_{D_i^S}^{Z|y,a}, P_{D_j^S}^{Z|y,a}\right)$ is derived in the similar way.

## E.2 PROOFS OF LEMMAS

**Proof of Lemma 6.** We have:

$$
\begin{aligned}
\int_{\mathcal{E}} \left| P_{D_j}^X - P_{D_i}^X \right| dX &= \int_{\mathcal{E}} \left( P_{D_j}^X - P_{D_i}^X \right) dX \\
&= \int_{\mathcal{E} \cup \overline{\mathcal{E}}} \left( P_{D_j}^X - P_{D_i}^X \right) dX - \int_{\overline{\mathcal{E}}} \left( P_{D_j}^X - P_{D_i}^X \right) dX \\
&= \int_{\overline{\mathcal{E}}} \left( P_{D_i}^X - P_{D_j}^X \right) dX \\
&= \int_{\overline{\mathcal{E}}} \left| P_{D_j}^X - P_{D_i}^X \right| dX \\
&= \frac{1}{2} \int \left| P_{D_j}^X - P_{D_i}^X \right| dX
\end{aligned}
$$

**Proof of Lemma 7.** We have:

$$
\begin{aligned}
\mathbb{E}_{D_j}[f(X)] = \int_{\mathcal{X}} f(X) P_{D_j}^X dX &= \int_{\mathcal{X}} f(X) P_{D_i}^X dX + \int_{\mathcal{X}} f(X) \left( P_{D_j}^X - P_{D_i}^X \right) dX \\
&= \mathbb{E}_{D_i}[f(X)] + \int_{\mathcal{X}} f(X) \left( P_{D_j}^X - P_{D_i}^X \right) dX \\
&= \mathbb{E}_{D_i}[f(X)] + \int_{\mathcal{E}} f(X) \left( P_{D_j}^X - P_{D_i}^X \right) dX + \int_{\overline{\mathcal{E}}} f(X) \left( P_{D_j}^X - P_{D_i}^X \right) dX \\
&\overset{(1)}{\leq} \mathbb{E}_{D_i}[f(X)] + \int_{\mathcal{E}} f(X) \left( P_{D_j}^X - P_{D_i}^X \right) dX \\
&\overset{(2)}{\leq} \mathbb{E}_{D_i}[f(X)] + C \int_{\mathcal{E}} \left( P_{D_j}^X - P_{D_i}^X \right) dX \\
&= \mathbb{E}_{D_i}[f(X)] + C \int_{\mathcal{E}} \left| P_{D_j}^X - P_{D_i}^X \right| dX \\
&\overset{(3)}{\leq} \mathbb{E}_{D_i}[f(X)] + \frac{C}{2} \int \left| P_{D_j}^X - P_{D_i}^X \right| dX \\
&\overset{(4)}{\leq} \mathbb{E}_{D_i}[f(X)] + \frac{C}{2} \sqrt{2 \min \left( \mathcal{D}_{KL} \left( P_{D_i}^X \parallel P_{D_j}^X \right), \mathcal{D}_{KL} \left( P_{D_j}^X \parallel P_{D_i}^X \right) \right)} \\
&= \mathbb{E}_{D_i}[f(X)] + \frac{C}{\sqrt{2}} \sqrt{\min \left( \mathcal{D}_{KL} \left( P_{D_i}^X \parallel P_{D_j}^X \right), \mathcal{D}_{KL} \left( P_{D_j}^X \parallel P_{D_i}^X \right) \right)}
\end{aligned}
$$

where $\mathcal{E}$ is the event that $P_{D_j}^X \geq P_{D_i}^X$ and $\overline{\mathcal{E}}$ is the complement of $\mathcal{E}$. We have $\overset{(1)}{\leq}$ because $\int_{\overline{\mathcal{E}}} f(X) \left( P_{D_j}^X - P_{D_i}^X \right) dX \leq 0$; $\overset{(2)}{\leq}$ because $f(X)$ is non-negative function and is bounded by $C$; $\overset{(3)}{\leq}$ by using Lemma 6; $\overset{(4)}{\leq}$ by using Pinsker's inequality between total variation norm and KL-divergence.

**Proof of Lemma 8.** Applying Lemma 7 and replacing $X$ by $(X, Y)$, $f$ by loss function $\mathcal{L}$, $D_i$ by $D_{i,j}$, we have:

$$
\begin{aligned}
\epsilon_{D_j}^{\text{Acc}}\left( \widehat{f} \right) - \mathbb{E}_{D_{i,j}} \left[ \mathcal{L}(\widehat{f}(X), Y) \right] &= \mathbb{E}_{D_j} \left[ \mathcal{L}(\widehat{f}(X), Y) \right] - \mathbb{E}_{D_{i,j}} \left[ \mathcal{L}(\widehat{f}(X), Y) \right] \\
&\leq \frac{C}{\sqrt{2}} \sqrt{\min \left( \mathcal{D}_{KL} \left( P_{D_j}^{X,Y} \parallel P_{D_{i,j}}^{X,Y} \right), \mathcal{D}_{KL} \left( P_{D_{i,j}}^{X,Y} \parallel P_{D_j}^{X,Y} \right) \right)} \\
&\leq \frac{C}{\sqrt{2}} \sqrt{\mathcal{D}_{KL} \left( P_{D_j}^{X,Y} \parallel P_{D_{i,j}}^{X,Y} \right)} \qquad (16)
\end{aligned}
$$

Applying Lemma 7 again and replacing $X$ by $(X, Y)$, $f$ by loss function $\mathcal{L}$, $D_j$ by $D_{i,j}$, we have:

$$
\begin{aligned}
\mathbb{E}_{D_{i,j}}\left[\mathcal{L}(\widehat{f}(X), Y)\right] - \epsilon_{D_i}^{\text{Acc}}\left(\widehat{f}\right) &= \mathbb{E}_{D_{i,j}}\left[\mathcal{L}(\widehat{f}(X), Y)\right] - \mathbb{E}_{D_i}\left[\mathcal{L}(\widehat{f}(X), Y)\right] \\
&\leq \frac{C}{\sqrt{2}}\sqrt{\min\left(\mathcal{D}_{KL}\left(P_{D_i}^{X,Y} \parallel P_{D_{i,j}}^{X,Y}\right), \mathcal{D}_{KL}\left(P_{D_{i,j}}^{X,Y} \parallel P_{D_i}^{X,Y}\right)\right)} \\
&\leq \frac{C}{\sqrt{2}}\sqrt{\mathcal{D}_{KL}\left(P_{D_i}^{X,Y} \parallel P_{D_{i,j}}^{X,Y}\right)} \qquad (17)
\end{aligned}
$$

Adding Eq. (16) to Eq. (17), we have:

$$
\begin{aligned}
\epsilon_{D_j}^{\text{Acc}}\left(\widehat{f}\right) - \epsilon_{D_i}^{\text{Acc}}\left(\widehat{f}\right) &\leq \frac{C}{\sqrt{2}}\left(\sqrt{\mathcal{D}_{KL}\left(P_{D_i}^{X,Y} \parallel P_{D_{i,j}}^{X,Y}\right)} + \sqrt{\mathcal{D}_{KL}\left(P_{D_j}^{X,Y} \parallel P_{D_{i,j}}^{X,Y}\right)}\right) \\
&\overset{(1)}{\leq} \frac{C}{\sqrt{2}}\sqrt{2\left(\mathcal{D}_{KL}\left(P_{D_i}^{X,Y} \parallel P_{D_{i,j}}^{X,Y}\right) + \mathcal{D}_{KL}\left(P_{D_j}^{X,Y} \parallel P_{D_{i,j}}^{X,Y}\right)\right)} \\
&= \frac{C}{\sqrt{2}}\sqrt{4\mathcal{D}_{JS}\left(P_{D_i}^{X,Y} \parallel P_{D_j}^{X,Y}\right)} \\
&= \sqrt{2}C d_{JS}\left(P_{D_i}^{X,Y}, P_{D_j}^{X,Y}\right)
\end{aligned}
$$

Here we have $\overset{(1)}{\leq}$ by using Cauchy–Schwarz inequality.

**Proof of Lemma 9.** Note that the JS-divergence $\mathcal{D}_{JS}\left(P_{D_i}^{X} \parallel P_{D_j}^{X}\right)$ can be understood as the mutual information between a random variable $X$ associated with the mixture distribution $P_{D_{i,j}}^{X} = \frac{1}{2}\left(P_{D_i}^{X} + P_{D_j}^{X}\right)$ and the equiprobable binary random variable $T$ used to switch between $P_{D_i}^{X}$ and $P_{D_j}^{X}$ to create the mixture distribution $P_{D_{i,j}}^{X}$. In particular, we have:

$$
\begin{aligned}
\mathcal{D}_{JS}\left(P_{D_i}^{X} \parallel P_{D_j}^{X}\right) &= \frac{1}{2}\left(\mathcal{D}_{KL}\left(P_{D_i}^{X} \parallel P_{D_{i,j}}^{X}\right) + \mathcal{D}_{JS}\left(P_{D_j}^{X} \parallel P_{D_{i,j}}^{X}\right)\right) \\
&= \frac{1}{2}\int\left(\log P_{D_i}^{X} - \log P_{D_{i,j}}^{X}\right)P_{D_i}^{X}dX \\
&\quad + \frac{1}{2}\int\left(\log P_{D_j}^{X} - \log P_{D_{i,j}}^{X}\right)P_{D_j}^{X}dX \\
&= \left(\frac{1}{2}\int\log\left(P_{D_i}^{X}\right)P_{D_i}^{X}dx + \frac{1}{2}\int\log\left(P_{D_j}^{X}\right)P_{D_j}^{X}dX\right) \\
&\quad - \int\log\left(P_{D_{i,j}}^{X}\right)P_{D_{i,j}}^{X}dX \\
&= -H(X|T) + H(X) \\
&= I(X;T)
\end{aligned}
$$

where $H(X)$ is the entropy of $X$, $H(X|T)$ is the entropy of $X$ conditioned on $T$, and $I(X;T)$ is the mutual information between $X$ and $T$. Similarly, we also have $\mathcal{D}_{JS}((P_{D_i}^{Z} \parallel P_{D_j}^{Z})) = I(Z;T)$. Because $Z$ is induced from $X$ by the mapping function $h$ then we have $Z \perp T \mid X$ and the Markov chain $T \to X \to Z$. According to data processing inequality for mutual information (Polyanskiy & Wu, 2014), we have $I(X;T) \geq I(Z;T)$ which implies $\mathcal{D}_{JS}((P_{D_i}^{X} \parallel P_{D_j}^{X})) \geq \mathcal{D}_{JS}((P_{D_i}^{Z} \parallel P_{D_j}^{Z}))$. Taking square root on both sides, we have $d_{JS}(P_{D_i}^{X}, P_{D_j}^{X}) \geq d_{JS}(P_{D_i}^{Z}, P_{D_j}^{Z})$.

**Proof of Lemma 10.**   We have:

$$
\epsilon_D^{\text{Acc}}\left(\widehat{h}\right) = \mathbb{E}_{z\sim P_D^Z, y\sim h_D(z)}\left[\mathcal{L}\left(\widehat{h}\left(Z\right), Y\right)\right]
$$

$$
= \sum_{y=1}^{|\mathcal{Y}|}\mathbb{E}_{z\sim P_D^Z}\left[\mathcal{L}\left(\widehat{h}\left(Z\right), y\right)h_D(Z)_y\right]
$$

$$
= \sum_{y=1}^{|\mathcal{Y}|}\int_{\mathcal{Z}}\mathcal{L}\left(\widehat{h}\left(Z\right), y\right)h_D(Z)_y P_D^Z dZ
$$

$$
= \sum_{y=1}^{|\mathcal{Y}|}\int_{\mathcal{Z}}\mathcal{L}\left(\widehat{h}\left(Z\right), y\right)\frac{\int_{g^{-1}(Z)} f_D(X)_y P_D^X dX}{\int_{g^{-1}(Z)} P_D^X dX}\int_{g^{-1}(Z)} P_D^X dX dZ
$$

$$
= \sum_{y=1}^{|\mathcal{Y}|}\int_{\mathcal{Z}}\mathcal{L}\left(\widehat{h}\left(Z\right), y\right)\int_{g^{-1}(Z)} f_D(X)_y P_D^X dX dZ
$$

$$
= \sum_{y=1}^{|\mathcal{Y}|}\int_{\mathcal{Z}}\int_{g^{-1}(Z)}\mathcal{L}\left(\widehat{h}\left(g(X)\right), y\right) f_D(X)_y P_D^X dX dZ
$$

$$
= \sum_{y=1}^{|\mathcal{Y}|}\int_{\mathcal{Z}}\int_{\mathcal{X}}\mathbb{1}\left(X\in g^{-1}(Z)\right)\mathcal{L}\left(\widehat{h}\left(Z\right), y\right) f_D(X)_y P_D^X dX dZ
$$

$$
= \sum_{y=1}^{|\mathcal{Y}|}\int_{\mathcal{X}}\int_{\mathcal{Z}}\mathbb{1}\left(Z = g(X)\right)\mathcal{L}\left(\widehat{h}\left(Z\right), y\right) f_D(X)_y P_D^X dX dZ
$$

$$
= \sum_{y=1}^{|\mathcal{Y}|}\int_{\mathcal{X}}\mathcal{L}\left(\widehat{h}\left(g(X)\right), y\right) f_D(X)_y P_D^X dX dZ
$$

$$
= \sum_{y=1}^{|\mathcal{Y}|}\int_{\mathcal{X}}\mathcal{L}\left(\widehat{f}\left(X\right), y\right) f_D(X)_y P_D^X dX
$$

$$
= \epsilon_D^{\text{Acc}}\left(\widehat{f}\right)
$$

**Proof of Lemma 11.**   We show the decomposition for KL-divergence first and then use the result to derive the decomposition for JS-divergence. We have:

$$
\mathcal{D}_{KL}\left(P_{D_i}^{X,Y}\ \|\ P_{D_j}^{X,Y}\right)
$$

$$
= \mathbb{E}_{D_i}\left[\log P_{D_i}^{X,Y} - \log P_{D_j}^{X,Y}\right]
$$

$$
= \mathbb{E}_{D_i}\left[\log P_{D_i}^Y + \log P_{D_i}^{X|Y}\right] - \mathbb{E}_{D_i}\left[\log P_{D_j}^Y + \log P_{D_j}^{X|Y}\right]
$$

$$
= \mathbb{E}_{D_i}\left[\log P_{D_i}^Y - \log P_{D_j}^Y\right] + \mathbb{E}_{D_i}\left[\log P_{D_i}^{X|Y} - \log P_{D_j}^{X|Y}\right]
$$

$$
= \mathbb{E}_{D_i}\left[\log P_{D_i}^Y - \log P_{D_j}^Y\right] + \mathbb{E}_{y\sim P_{D_i}^Y}\left[\mathbb{E}_{x\sim P_{D_i}^{X|y}}\left[\log P_{D_i}^{X|Y} - \log P_{D_j}^{X|Y}\right]\right]
$$

$$
= \mathcal{D}_{KL}\left(P_{D_i}^Y\ \|\ P_{D_j}^Y\right) + \mathbb{E}_{D_i}\left[\mathcal{D}_{KL}\left(P_{D_i}^{X|Y}\ \|\ P_{D_j}^{X|Y}\right)\right]
$$

$$\mathcal{D}_{JS}\left(P_{D_i}^{X,Y} \parallel P_{D_j}^{X,Y}\right)$$

$$= \frac{1}{2}\left(\mathcal{D}_{KL}\left(P_{D_i}^{X,Y} \parallel P_{D_{i,j}}^{X,Y}\right)\right) + \frac{1}{2}\left(\mathcal{D}_{KL}\left(P_{D_j}^{X,Y} \parallel P_{D_{i,j}}^{X,Y}\right)\right)$$

$$= \frac{1}{2}\left(\mathcal{D}_{KL}\left(P_{D_i}^{Y} \parallel P_{D_{i,j}}^{Y}\right)\right) + \frac{1}{2}\left(\mathbb{E}_{D_i}\left[\mathcal{D}_{KL}\left(P_{D_i}^{X|Y} \parallel P_{D_{i,j}}^{X|Y}\right)\right]\right)$$

$$+ \frac{1}{2}\left(\mathcal{D}_{KL}\left(P_{D_j}^{Y} \parallel P_{D_{i,j}}^{Y}\right)\right) + \frac{1}{2}\left(\mathbb{E}_{D_j}\left[\mathcal{D}_{KL}\left(P_{D_j}^{X|Y} \parallel P_{D_{i,j}}^{X|Y}\right)\right]\right)$$

$$= \mathcal{D}_{JS}\left(P_{D_i}^{Y} \parallel P_{D_j}^{Y}\right) + \frac{1}{2}\left(\mathbb{E}_{D_i}\left[\mathcal{D}_{KL}\left(P_{D_i}^{X|Y} \parallel P_{D_{i,j}}^{X|Y}\right)\right]\right) + \frac{1}{2}\left(\mathbb{E}_{D_j}\left[\mathcal{D}_{KL}\left(P_{D_j}^{X|Y} \parallel P_{D_{i,j}}^{X|Y}\right)\right]\right)$$

$$\leq \mathcal{D}_{JS}\left(P_{D_i}^{Y} \parallel P_{D_j}^{Y}\right) + \frac{1}{2}\left(\mathbb{E}_{D_i}\left[\mathcal{D}_{KL}\left(P_{D_i}^{X|Y} \parallel P_{D_{i,j}}^{X|Y}\right)\right]\right) + \frac{1}{2}\left(\mathbb{E}_{D_i}\left[\mathcal{D}_{KL}\left(P_{D_j}^{X|Y} \parallel P_{D_{i,j}}^{X|Y}\right)\right]\right)$$

$$+ \frac{1}{2}\left(\mathbb{E}_{D_j}\left[\mathcal{D}_{KL}\left(P_{D_j}^{X|Y} \parallel P_{D_{i,j}}^{X|Y}\right)\right]\right) + \frac{1}{2}\left(\mathbb{E}_{D_j}\left[\mathcal{D}_{KL}\left(P_{D_i}^{X|Y} \parallel P_{D_{i,j}}^{X|Y}\right)\right]\right)$$

$$= \mathcal{D}_{JS}\left(P_{D_i}^{Y} \parallel P_{D_j}^{Y}\right) + \mathbb{E}_{D_i}\left[\mathcal{D}_{JS}\left(P_{D_i}^{X|Y} \parallel P_{D_j}^{X|Y}\right)\right] + \mathbb{E}_{D_j}\left[\mathcal{D}_{JS}\left(P_{D_i}^{X|Y} \parallel P_{D_j}^{X|Y}\right)\right]$$

**Proof of Lemma 12.**

$$\begin{aligned}
\mathbb{E}_D\left[\mathcal{L}(\widehat{f}(X), Y)\right] &= \mathbb{E}_D\left[\sum_{\widehat{y}\in\mathcal{Y}} \widehat{f}(X)_{\widehat{y}} L(\widehat{y}, Y)\right] \\
&\overset{(1)}{\geq} c\,\mathbb{E}_X\left[\sum_{\widehat{y}\in\mathcal{Y}} \widehat{f}(X)_{\widehat{y}} \Pr(Y \neq \widehat{y}|X)\right] \\
&\overset{(2)}{=} c\,\mathbb{E}_X\left[1 - \widehat{f}(X)^T f(X)\right] \\
&\overset{(3)}{\geq} \frac{c}{2}\,\mathbb{E}_X\left[\left\|\widehat{f}(X) - f(X)\right\|_2^2\right] \\
&\overset{(4)}{\geq} \frac{c}{2}\frac{1}{|\mathcal{Y}|}\mathbb{E}_X\left[\left(\left\|\widehat{f}(X) - f(X)\right\|_1\right)^2\right] \\
&\overset{(5)}{\geq} \frac{c}{2}\frac{1}{|\mathcal{Y}|}\left(\left\|\mathbb{E}_X\left[\widehat{f}(X) - f(X)\right]\right\|_1\right)^2 \\
&= \frac{c}{2}\frac{1}{|\mathcal{Y}|}\left\|P_D^{\widehat{Y}} - P_D^{Y}\right\|_1^2 \\
&\overset{(6)}{\geq} \frac{2c}{|\mathcal{Y}|}\mathcal{D}_{JS}\left(P_D^{Y} \parallel P_D^{\widehat{Y}}\right)^2 \\
&= \frac{2c}{|\mathcal{Y}|}\cdot d_{JS}\left(P_D^{Y}, P_D^{\widehat{Y}}\right)^4
\end{aligned}$$

Here we have $\overset{(1)}{\geq}$ is because of the assumption that $L(\widehat{y}, y)$ is lower bounded by $c$ when $\widehat{y} \neq y$; $\overset{(2)}{=}$ is because $\widehat{f}(X)^T \mathbf{1} = ||\widehat{f}(X)||_1 = 1$; $\overset{(3)}{\geq}$ is because $||\widehat{f}(X)||_2 \leq ||\widehat{f}(X)||_1 = 1$; $\overset{(4)}{\geq}$ is because $||\widehat{f}(X)||_2 \geq \frac{1}{\sqrt{|\mathcal{Y}|}}||\widehat{f}(X)||_1$; $\overset{(5)}{\geq}$ is by using Jensen's inequality; $\overset{(6)}{\geq}$ is by using JS-divergence lower bound of total variation distance.

**Proof of Lemma 14.** Similar to the proof in Lemma 8, we apply Lemma 7 for $R_{D_i}^{y,a}$ and $R_{D_j}^{y,a}$ and note that $\widehat{f}(X)_y$ is bounded by 1. Then $\forall y, a \in \{0, 1\}$, we have:

$$R_{D_j}^{y,a} - \mathbb{E}_{D_{i,j}} \left[ \widehat{f}(X)_y | Y = y, A = a \right]$$

$$= \mathbb{E}_{D_j} \left[ \widehat{f}(X)_y | Y = y, A = a \right] - \mathbb{E}_{D_{i,j}} \left[ \widehat{f}(X)_y | Y = y, A = a \right]$$

$$\leq \frac{1}{\sqrt{2}} \sqrt{\min \left( \mathcal{D}_{KL} \left( P_{D_i}^{X|Y=y,A=a} \parallel P_{D_{i,j}}^{X|Y=y,A=a} \right), \mathcal{D}_{KL} \left( P_{D_{i,j}}^{X|Y=y,A=a} \parallel P_{D_i}^{X|Y=y,A=a} \right) \right)}$$

$$\leq \frac{1}{\sqrt{2}} \sqrt{\mathcal{D}_{KL} \left( P_{D_j}^{X|Y=y,A=a} \parallel P_{D_{i,j}}^{X|Y=y,A=a} \right)} \tag{18}$$

$$\mathbb{E}_{D_{i,j}} \left[ \widehat{f}(X)_y | Y = y, A = a \right] - R_{D_i}^{y,a}$$

$$= \mathbb{E}_{D_{i,j}} \left[ \widehat{f}(X)_y | Y = y, A = a \right] - \mathbb{E}_{D_i} \left[ \widehat{f}(X)_y | Y = y, A = a \right]$$

$$\leq \frac{1}{\sqrt{2}} \sqrt{\min \left( \mathcal{D}_{KL} \left( P_{D_j}^{X|Y=y,A=a} \parallel P_{D_{i,j}}^{X|Y=y,A=a} \right), \mathcal{D}_{KL} \left( P_{D_{i,j}}^{X|Y=y,A=a} \parallel P_{D_j}^{X|Y=y,A=a} \right) \right)}$$

$$\leq \frac{1}{\sqrt{2}} \sqrt{\mathcal{D}_{KL} \left( P_{D_i}^{X|Y=y,A=a} \parallel P_{D_{i,j}}^{X|Y=y,A=a} \right)} \tag{19}$$

Adding Eq. (18) to Eq. (19), we have:

$$R_{D_j}^{y,a} - R_{D_i}^{y,a}$$

$$\leq \frac{1}{\sqrt{2}} \left( \sqrt{\mathcal{D}_{KL} \left( P_{D_i}^{X|Y=y,A=a} \parallel P_{D_{i,j}}^{X|Y=y,A=a} \right)} + \sqrt{\mathcal{D}_{KL} \left( P_{D_j}^{X|Y=y,A=a} \parallel P_{D_{i,j}}^{X|Y=y,A=a} \right)} \right)$$

$$\leq \sqrt{2} d_{JS} \left( P_{D_j}^{X|Y=y,A=a}, P_{D_{i,j}}^{X|Y=y,A=a} \right)$$

**Proof of Lemma 15.** We give the proof for unfairness measure w.r.t. to equal opportunity first and then use this result to derive the proof for unfairness measure w.r.t. to equalized odd. Without loss of generality, assign group indices $1, 0$ be such that $R_{D_j}^{1,0} \left( \widehat{f} \right) \geq R_{D_j}^{1,1} \left( \widehat{f} \right)$. Then we have:

$$\epsilon_{D_j}^{\text{EP}} \left( \widehat{f} \right) = \left| R_{D_j}^{1,0} \left( \widehat{f} \right) - R_{D_j}^{1,1} \left( \widehat{f} \right) \right|$$

$$= R_{D_j}^{1,0} \left( \widehat{f} \right) - R_{D_j}^{1,1} \left( \widehat{f} \right)$$

$$= R_{D_j}^{1,0} \left( \widehat{f} \right) - \mathbb{E}_{D_j} \left[ \widehat{f}(X)_1 | Y = 1, A = 1 \right]$$

$$= R_{D_j}^{1,0} \left( \widehat{f} \right) + \mathbb{E}_{D_j} \left[ 1 - \widehat{f}(X)_1 | Y = 1, A = 1 \right] - 1$$

$$= R_{D_j}^{1,0} \left( \widehat{f} \right) + R_{D_j}^{1,1} \left( \mathbf{1} - \widehat{f} \right) - 1$$

where $\mathbf{1}$ is vector with all 1's. By Lemma 14, we have:

$$R_{D_j}^{1,0} \left( \widehat{f} \right) \leq R_{D_i}^{1,0} \left( \widehat{f} \right) + \sqrt{2} d_{JS} \left( P_{D_j}^{X|Y=1,A=0}, P_{D_i}^{X|Y=1,A=0} \right)$$

$$R_{D_j}^{1,1} \left( \mathbf{1} - \widehat{f} \right) \leq R_{D_i}^{1,1} \left( \mathbf{1} - \widehat{f} \right) + \sqrt{2} d_{JS} \left( P_{D_j}^{X|Y=1,A=1}, P_{D_i}^{X|Y=1,A=1} \right)$$

Sum above two inequalities and add $-1$ at both sides, we have,

$$\epsilon_{D_j}^{\text{EP}} \left( \widehat{f} \right) = R_{D_j}^{1,0} \left( \widehat{f} \right) + R_{D_j}^{1,1} \left( \mathbf{1} - \widehat{f} \right) - 1$$

$$\leq R_{D_i}^{1,0} \left( \widehat{f} \right) + R_{D_i}^{1,1} \left( \mathbf{1} - \widehat{f} \right) - 1 + \sqrt{2} \sum_{a=0,1} d_{JS} \left( P_{D_j}^{X|Y=1,A=a}, P_{D_i}^{X|Y=1,A=a} \right)$$

$$\leq \epsilon_{D_i}^{\text{EP}} \left( \widehat{f} \right) + \sqrt{2} \sum_{a=0,1} d_{JS} \left( P_{D_j}^{X|Y=1,A=a}, P_{D_i}^{X|Y=1,A=a} \right) \tag{20}$$

Similarly, we have:

$$\left| R_{D_j}^{0,0}\left(\widehat{f}\right) - R_{D_j}^{0,1}\left(\widehat{f}\right) \right| \le \left| R_{D_i}^{0,0}\left(\widehat{f}\right) - R_{D_i}^{0,1}\left(\widehat{f}\right) \right| + \sqrt{2} \sum_{a=0,1} d_{JS}\left( P_{D_j}^{X|Y=0,A=a}, P_{D_i}^{X|Y=0,A=a} \right)$$

(21)

Sum both Eq. (20) and Eq. (21), we have:

$$\epsilon_{D_j}^{\text{EO}}\left(\widehat{f}\right) \le \epsilon_{D_i}^{\text{EO}}\left(\widehat{f}\right) + \sqrt{2} \sum_{y=0,1} \sum_{a=0,1} d_{JS}\left( P_{D_j}^{X|Y=y,A=a}, P_{D_i}^{X|Y=y,A=a} \right)$$

**Proof of Lemma 16.**  Similar to the proof of Lemma 10, $R_{D_i}^{y,a}\left(\widehat{f}\right) = R_{D_i}^{y,a}\left(\widehat{h}\right) \; \forall y, a \in \{0,1\}$. Then, we have:

$$\begin{aligned}
\epsilon_{D_i}^{\text{EO}}\left(\widehat{f}\right) &= \left| R_{D_i}^{0,0}\left(\widehat{f}\right) - R_{D_i}^{0,1}\left(\widehat{f}\right) \right| + \left| R_{D_i}^{1,0}\left(\widehat{f}\right) - R_{D_i}^{1,1}\left(\widehat{f}\right) \right| \\
&= \left| R_{D_i}^{0,0}\left(\widehat{h}\right) - R_{D_i}^{0,1}\left(\widehat{h}\right) \right| + \left| R_{D_i}^{1,0}\left(\widehat{h}\right) - R_{D_i}^{1,1}\left(\widehat{h}\right) \right| \\
&= \epsilon_{D_i}^{\text{EO}}\left(\widehat{h}\right) \\
\epsilon_{D_i}^{\text{EP}}\left(\widehat{f}\right) &= \left| R_{D_i}^{1,0}\left(\widehat{f}\right) - R_{D_i}^{1,1}\left(\widehat{f}\right) \right| \\
&= \left| R_{D_i}^{1,0}\left(\widehat{h}\right) - R_{D_i}^{1,1}\left(\widehat{h}\right) \right| \\
&= \epsilon_{D_i}^{\text{EP}}\left(\widehat{h}\right)
\end{aligned}$$

**Proof of Lemma 17.**  $\forall y \in \mathcal{Y}$, we have:

$$\begin{aligned}
\mathcal{D}_{JS}\left( P_i^{Z|y} \parallel P_j^{Z|y} \right) &\overset{(1)}{=} \mathcal{D}_{JS}\left( \int_{\mathcal{X}} P^{Z|x} P_i^{x|y} dx \parallel \int_{\mathcal{X}} P^{Z|m_{i,j}^y(x)} P_j^{m_{i,j}^y(x)|y} dm_{i,j}^y(x) \right) \\
&\overset{(2)}{=} \mathcal{D}_{JS}\left( \int_{\mathcal{X}} P^{Z|x} P_i^{x|y} dx \parallel \int_{\mathcal{X}} P^{Z|m_{i,j}^y(x)} P_j^{m_{i,j}^y(x)|y} dx \right) \\
&\overset{(3)}{=} \mathcal{D}_{JS}\left( \int_{\mathcal{X}} P^{Z|x} P_i^{x|y} dx \parallel \int_{\mathcal{X}} P^{Z|m_{i,j}^y(x)} P_i^{x|y} dx \right) \\
&\overset{(4)}{\le} \int_{\mathcal{X}} P_i^{x|y} \mathcal{D}_{JS}\left( P^{Z|x} \parallel P^{Z|m_{i,j}^y(x)} \right) dx
\end{aligned}$$

Here we have $\overset{(1)}{=}$ is because of law of total probability and $Z \perp Y|X$; $\overset{(2)}{=}$ is because $m_{i,j}^y$ is invertible function; $\overset{(3)}{=}$ is because $P_i^{x|y} = P_j^{m_{i,j}^y(x)|y} \; \forall x \in \mathcal{X}$; $\overset{(4)}{\le}$ is because of joint complexity of JS divergence. By similar derivation, $\forall y \in \mathcal{Y}, a \in \mathcal{A}$, we have:

$$\mathcal{D}_{JS}\left( P_i^{Z|y,a} \parallel P_j^{Z|y,a} \right) \le \int_{\mathcal{X}} P_i^{x|y,a} \mathcal{D}_{JS}\left( P^{Z|x} \parallel P^{Z|m_{i,j}^{y,a}(x)} \right) dx$$

