# OpenReview forum: "Fairness and Accuracy under Domain Generalization"
_ICLR.cc/2023/Conference — ICLR 2023 poster_

### Official Review · Reviewer_dQPF · 2022-10-24

**Confidence:** 3
**Correctness:** 4
**Technical Novelty And Significance:** 2
**Empirical Novelty And Significance:** 3
**Recommendation:** 6

**Clarity, Quality, Novelty And Reproducibility:**

The paper is well-written. It can be improved, however, if some of the notations and concepts can be introduced within a context. The theoretical results are interesting because they provide guidance on designing better algorithms.

**Strength And Weaknesses:**

In general, this paper is well-written. The presentation is clear, references are well-cited and motivation is sufficiently discussed. The upper and lower bounds on accuracy and fairness transferability are interesting because they serve as good guidance for algorithm design. Below I have some comments:
1. Section 2 is a bit dense with too many notations and concepts, which may be hard for readers not super familiar with the terminologies. It would be better if some of the definitions/terminologies can be introduced within a context.
2. On page 3, it is clear what $\mathcal{H}$ and $\mathcal{H}\Delta \mathcal{H}$ are (they seem to be defined in the Appendix). Maybe add a reference here.
3. On page 4, the footnote mark 2 and 3 should be put after punctuations.
4. Above Theorem 2, "bellow" -> "below"
5. On page 5, maybe add a sentence or two to explain the claim "Note that these results are consistent with Thm. 1 and Thm. 3 above."
6. I assume a similar lower bound as Thm 2 can be established for EO as well?


**Summary Of The Paper:**

This paper considers the problem of domain generalization with both fairness and accuracy guarantees. To this end, the authors establish an upper bound for accuracy (resp. fairness) for general domain generalization algorithms, which in turn provides guidance on designing their proposed algorithm. A complementary lower bound is also provided in the paper, explaining why previous representation learning methods may not be good choices. The proposed method is tested on real data, showing substantial improvement over previous practices.

**Summary Of The Review:**

In general, this is a well-written paper, with a clear presentation and sufficient discussion on previous works. The theory presented here is novel and interesting; the advantage of the proposed algorithm is validated by numerical experiments.

---

> ### Author Response · Authors · 2022-11-16
> **Response to Reviewer dQPF**
>
>  Thank you for your constructive review. We appreciate that you believe our theoretical findings are novel and interesting, and our proposed algorithm is well-motivated. We would like to address your questions as follows. The manuscript and supplementary are also revised and are highlighted in red.
>
> > **Q1:** Section 2 is a bit dense with too many notations and concepts, which may be hard for readers not super familiar with the terminologies. It would be better if some of the definitions/terminologies can be introduced within a context.
>
>  **A1:** Thanks for your suggestion. We’ve added the contexts when introducing the definitions/terminologies in our revised manuscript.
>
> > **Q2:** On page 3, it is clear what $\mathcal{H}$ and $\mathcal{H}\Delta\mathcal{H}$ are (they seem to be defined in the Appendix). Maybe add a reference here.
>
>  **A2:** Thanks for your suggestion. We’ve added references to our revised manuscript.
>
> > **Q3:**  On page 4, the footnote mark 2 and 3 should be put after punctuations. Above Theorem 2, "bellow" -> "below"
>
>  **A3:** Thanks for pointing out the typos. We’ve fixed them in our revised manuscript.
>
> >  **Q4:** On page 5, maybe add a sentence or two to explain the claim "Note that these results are consistent with Thm. 1 and Thm. 3 above."
>
>  **A4:** Thanks for your suggestion. We’ve added more information for this claim in our revised manuscript.
>
> >  **Q5:** I assume a similar lower bound as Thm 2 can be established for EO as well?
>
>  **A5:**  The lower bound for unfairness (i.e., equalized odds and equal opportunity) is zero because we can always achieve perfectly fair predictions by making the prediction $\widehat{f}(X)$ to be independent with label $Y$ and sensitive attribute $A$. However, such a random classifier is inaccurate and is not our objective. Therefore, it is not necessary to develop the lower bound for unfairness.

---

### Official Review · Reviewer_PEwK · 2022-10-25

**Confidence:** 4
**Correctness:** 3
**Technical Novelty And Significance:** 3
**Empirical Novelty And Significance:** 3
**Recommendation:** 6

**Clarity, Quality, Novelty And Reproducibility:**

Overall, I found the paper not easy to follow. It would be greatly appreciated if contents can be organized to better illustrate the implication of results, especially in the part where notations are dense (e.g., Section 4).

The proposed algorithm is novel (to the best of my knowledge). The paper also specifies the technical details for reproducing the experimental results, as well as additional experiments in the appendix.

**Strength And Weaknesses:**

## Strength

The paper takes effort to make theoretical results mathematically rigorous and provides multiple bounds w.r.t. accuracy and fairness. An empirical two-stage algorithm is also presented for the purpose of learning invariant representation.

## Weakness

### 1. w.r.t. dense notations and clarity of presentation

Overall, while I appreciate the effort of presenting rigorous arguments, I find the paper hard to follow from time to time because of the dense notations and (occasionally unnecessarily) complex presentation. For example, in Section 4, the overall optimizing objective, the description of the proposed two-stage algorithm, the calculation of density of learnt representation are not presented in a clearly organized way. Personally, I would benefit a lot from, e.g., the numbering of the key equation (e.g., the objective), the implication and/or importance of the theoretical results, and the correspondence between the specific step in algorithm to previous theoretical results. As another example, the sufficient conditions provided in Theorem 4 seem to be rather straightforward, I think a concise summary would also serve the purpose without sacrificing mathematical rigor.

### 2. the relation between the proposed FATDM algorithm and previous theoretical results

While I appreciate the mathematical rigor when deriving upper-/lower- bounds for accuracy and upper bound for fairness. I am having some difficulty drawing connection between the theoretical results and the empirical two-stage algorithm. Does the theoretical characterization provides the ground for framing the problem in terms of invariant representation learning? Or, does the theoretical analysis motivate the design of algorithm? After all, as authors already noted, in the domain generalization setting, we do not have access to target domain, in this case, I am wondering if authors can share some insight regarding how to parse the theoretical results in the evaluation of empirical algorithm.

## Additional comment

### w.r.t. the theoretical analysis on Fairness transferability

The paper discusses previous approaches in extensive detail, which is clear and informative. Only for the purpose of completeness, I would like to provide a pointer to a previous work that also provided fairness transferability bounds for DP and EO notions: "Fairness Transferability Subject to Bounded Distribution Shift" by Chen et al., (2022). I am aware of the fact that the paper is recent (arxiv June 2022, to appear in NeurIPS), but considering that they are considering the same fairness notions in a very similar problem setting -- fairness bounds for DP and EO under domain shift, I want to bring their paper to the attention of authors. I am just a little bit hesitant to agree with the claim that the paper is "the first work studying domain generalization with fairness consideration". In fact, their bounds are more general, e.g., they do not require $A$ and $Y$ to be binary for EO when deriving the fairness bound.




**Summary Of The Paper:**

The paper considers the representation learning problem in domain generalization setting with fairness constraints (DP, EO). The paper presents upper and lower bounds for accuracy, and also an upper bound for fairness violation. Empirically, the paper aims to solve a minmax problem to account for fairness when learning invariant representations.

**Summary Of The Review:**

As presented in the paper, the bounds for accuracy when there are domain shifts have been discussed extensively. Fairness bounds are actually also discussed in some recent literature (as I mentioned in "Strength and Weakness"). I think largely the contribution of the paper comes from the empirical invariant representation learning algorithm. Additional clarifications of the connection between theoretical and empirical results would be very helpful.

=== Post-rebuttal ===

Thank authors for the additional discussions and clarifications. I have updated my score accordingly.

---

> ### Author Response · Authors · 2022-11-16
> **Response to Reviewer PEwK**
>
> Thank you for your constructive review. We appreciate that you believe our work is novel. We would like to address your question as follows. The manuscript and supplementary are also revised and we highlight the changes in red.
>
> > **Q1:** w.r.t. dense notations and clarity of presentation
>
> **A1:** Thanks for your suggestions. We have made the following revisions to improve the presentation: (1) we highlighted the key equations; (2) we added a remark at the end of section 4 to summarize the implications of our theoretical results, and the connection between theory and the proposed algorithm; (3) we revised Thm. 4 and make it more concise.
>
> > **Q2:** the relation between the proposed FATDM algorithm and previous theoretical results
>
>  **A2:** Thm. 1 and Thm. 3 state the upper bounds for error and unfairness in the unseen target domain. They suggest a way to ensure high accuracy and fairness in target domains: by minimizing the source error $\epsilon_{D^s_i}^{Acc}$ (i.e., $L_{cls}$ in Eq. (3)), the source unfairness $\epsilon_{D^s_i}^{EO}$ (i.e., $L_{fair}$ in Eq. (3)), and the discrepancies between source domains in representation space $d_{JS}\left(P_{D_i^S}^{Z|Y=y},P_{D_j^S}^{Z|Y=y}\right)$ and $d_{JS}\left(P_{D_i^S}^{Z|Y=y,A=a},P_{D_j^S}^{Z|Y=y,A=a}\right)$ (i.e., $L_{inv}$ in Eq. (3)). The usages of these loss terms in Eq. (3) are summarized in Table 1.
>
> The common approach to optimizing Eq. (3) is based on adversarial learning (as shown in Eq. (4)), which is not stable when $|\mathcal{Y} \times \mathcal{A}|$ is large. In contrast, FATDM addresses this issue via a two-stage learning framework motivated by Thm. 5. Specifically, Eq. (3) can be optimized by: (i) finding mapping functions $m_{i,j}^{y}$ and $m_{i,j}^{y,a}$ (procedure **Density_Matching** in Alg. 1) and (ii) minimizing Eq. (3) with $L_{inv}$ defined in Eq. (5) (procedure **Invariant_Representation_Learning** in Alg. 1). This explanation is also added to section 4 in our revised manuscript (remark 2).
>
>
> > **Q3:** w.r.t. the theoretical analysis on Fairness transferability
>
> **A3:** Thanks for pointing out the interesting concurrent work (Chen et al., 2022). We’ve cited this in the related work section (Appendix A). To further support our claim “the first work studying domain generalization with fairness consideration,” we would like to clarify as follows.
>
> 1. (Chen et al., 2022) studied the transfer of fairness under domain adaptation which requires the information of target domain at training time (i.e., the bounded group-vectorized shift/distance between source and target domains in this paper). The setting of domain adaptation has also been explored in some other recent works, as we discussed in Appendix A. In contrast, our work focuses on fairness transfer under domain generalization when the target domain data are inaccessible during training; it instead assumes that there exists a set of source domains based on which the learned model can be generalized to an unseen, novel target domain.
>
> 2. Indeed, our theoretical bound for fairness is not limited to binary classification setting. As mentioned in section 2 (unfairness metric), our bound can be easily generalized to multi-class, multi-protected attribute setting: by replacing the sum notations $\sum_{y \in \{ 0, 1 \}}$ and $\sum_{a \in \{ 0, 1 \}}$ in the upper bound in Thm. 3 with $\sum_{y \in \mathcal{Y}}$ and $\sum_{a \in \mathcal{A} }$.

---

### Official Review · Reviewer_nacW · 2022-10-27

**Confidence:** 3
**Clarity, Quality, Novelty And Reproducibility:** The paper is clearly written, and the…
**Correctness:** 4
**Technical Novelty And Significance:** 4
**Empirical Novelty And Significance:** 3
**Recommendation:** 8

**Strength And Weaknesses:**

Strength And Weaknesses:
[Strength]

S1: The paper tries to demonstrate how to achieve fair and accurate predictions in unseen environments without relying on assumptions about a specific type of distribution shifts across domains, which is an important and practical issue.

S2: The theoretical analyses in the paper give several nice insights. For example, I enjoy reading Sections 3 and 4.

S3: The proposed algorithm is effective in specific empirical scenarios, as shown in their theoretical results.

[Weakness]

Adding discussion about how the tradeoff between accuracy and fairness is affected by the number of source domains would be helpful. A major issue in certain settings could be acquiring multiple domain data, in which case one is restricted to a single domain. It would be helpful to address this point.


**Summary Of The Paper:**

The paper aims to develop fair and accurate models in unseen environments. The use of multiple environments and invariant representation learning for training fair models is an interesting approach. The paper also provides a theoretical analysis on the efficiency of invariant representation learning in transferring fair and accurate models across domains while using multiple source domains. The empirical results support the theoretical claims.




**Summary Of The Review:**

The paper tries to solve the important issue of achieving fair and accurate predictions in unseen environments. As the paper gives several insights into their theoretical discussion, I vote for accepting this paper.

---

> ### Author Response · Authors · 2022-11-16
> **Response to Reviewer nacW**
>
> Thank you for your positive review. We appreciate that you believe our work is important and practical. We would like to address your question as follows.
>
> > **Q1:** Adding discussion about how the tradeoff between accuracy and fairness is affected by the number of source domains would be helpful.
>
> **A1:** Thanks for your suggestion. We conducted an additional experiment to investigate the relationship between the fairness-accuracy tradeoff on unseen target domains and the number of source domains during training; the experimental result and the corresponding discussion are included in **Appendix C** in the revised manuscript.
>
> In this experiment, we evaluate the performances of **FATDM** and **ERM** on Edema dataset with different numbers of source domains.  We first construct the dataset for each domain by rotating images with $\theta$ degree, where $\theta \in \\{ 0^{\circ}, 15^{\circ}, 30^{\circ}  \\}$ when the number of domains is $3$, $\theta \in \\{ 0^{\circ}, 15^{\circ}, 30^{\circ},45^{\circ}\\}$ when the number of domains is $4$, and $\theta \in \\{ 0^{\circ}, 15^{\circ}, 30^{\circ},45^{\circ}, 60^{\circ}\\}$ when the number of domains is $5$. The number of images per domain is adapted to ensure the training set size is fixed for the three cases. We follow the leave-one-out domain setting in which one domain serves as the unseen target domain for evaluation while the rest domains are for training; the average results across target domains are reported.
>
> Figure 11 in Appendix C shows error-unfairness curves of **FATDM** and **ERM** when training with $2$, $3$, and $4$ source domains. We observe that training with more source domains does not always help the model achieve a better fairness-accuracy tradeoff on unseen target domains. In particular, the performances of both **FATDM** and **ERM** are the best when training with 2 source domains and the worst when training with 3 source domains. We conjecture the reason that adding more source domains may help reduce the discrepancy between source and target domains (term (ii) in Thm. 1 and Thm. 3), but it may make it more difficult to minimize the source error and unfairness (term (i) in Thm. 1 and Thm. 3) and to learn invariant representation across the source domains (term (iii) in Thm. 1 and Thm. 3). Thus, our suggestion in practice is to conduct an ablation study to find the optimal number of source domains.
>
> > **Q2:** A major issue in certain settings could be acquiring multiple domain data, in which case one is restricted to a single domain.
>
> **A2:** We clarify that our work focuses on transferring fairness and accuracy under **domain generalization** when the target domain data are inaccessible during training. Instead, it relies on a set of multiple source domains to generalize to an unseen, novel target domain. The case of a single domain requires some information about the target domain to perform fairness and accuracy transfer. This setting belongs to **domain adaptation** which has been explored before as shown in Appendix A in our manuscript.

---

### Decision · Program_Chairs · 2023-01-20

**Decision:**

Accept: poster

**Justification For Why Not Higher Score:**

Too technical with complex notations.

**Justification For Why Not Lower Score:**

NA

**Metareview: Summary, Strengths And Weaknesses:**

Domain generalization (DG) is a core problem in deep learning, and it has attracted much attention in the past few years.  While previous DG methods are concerned only with out-of-distribution (OOD) classification performance, this paper also considers fairness measures (equalized odds). The authors present some theoretical results regarding the conditions under which fairness and accuracy can be perfectly transferred OOD via invariant representation learning, and propose a learning algorithm to achieve good OOD classification and fairness performance.   The reviewers find the theoretical results interesting, the algorithm novel, and the empirical evaluation satisfactory, although there is a feeling that the math notations should be improved (complex superscript + subscript&superscript of subscript).

**Note From Pc:**

if the above contains the word "oral" or "spotlight" please see: "oral" presentation means -> notable-top-5% and "spotlight" means -> notable-top-25%. As stated in our emails, we are disassociating presentation type from AC recommendations